# Is Homophily a Necessity for Graph Neural Networks?

**Yao Ma**
New Jersey Institute of Technology
yao.ma@njit.edu

**Xiaorui Liu**
Michigan State University
xiaorui@msu.edu

**Neil Shah**
Snap Inc.
nshah@snap.com

**Jiliang Tang**
Michigan State University
tangjili@msu.edu

## Abstract

Graph neural networks (GNNs) have shown great prowess in learning representations suitable for numerous graph-based machine learning tasks. When applied to semi-supervised node classification, GNNs are widely believed to work well due to the homophily assumption ("like attracts like"), and fail to generalize to heterophilous graphs where dissimilar nodes connect. Recent works have designed new architectures to overcome such heterophily-related limitations. However, we empirically find that standard graph convolutional networks (GCNs) can actually achieve strong performance on some commonly used heterophilous graphs. This motivates us to reconsider whether homophily is truly necessary for good GNN performance. We find that this claim is not *quite* accurate, and certain types of "good" heterophily exist, under which GCNs can achieve strong performance. Our work carefully characterizes the implications of different heterophily conditions, and provides supporting theoretical understanding and empirical observations. Finally, we examine existing heterophilous graphs benchmarks and reconcile how the GCN (under)performs on them based on this understanding.

## 1 Introduction

Graph neural networks (GNNs) are a prominent approach for learning representations for graph structured data. Thanks to their great capacity in jointly leveraging attribute and graph structure information, they have been widely adopted to promote improvements for numerous graph-related learning tasks (Kipf and Welling, 2016; Hamilton et al., 2017; Ying et al., 2018; Fan et al., 2019; Zitnik et al., 2018), especially centered around node representation learning and semi-supervised node classification (SSNC). GNNs learn node representations by a recursive neighborhood aggregation process, where each node aggregates and transforms features from its neighbors. The node representations can then be utilized for downstream node classification or regression tasks. Due to this neighborhood aggregation mechanism, several existing works posit that many GNNs implicitly assume strong homophily and homophily is critical for GNNs to achieve strong performance on SSNC (Zhu et al., 2020b;a; Chien et al., 2021; Maurya et al., 2021; Halcrow et al., 2020; Lim et al., 2021). In general, homophily describes the

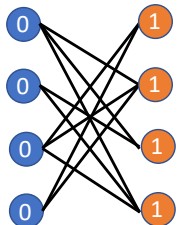

Figure 1: A heterophilous graph on which GCN achieves perfect performance.

phenomenon that nodes tend to connect with "similar" or "alike" others. Homophily is observed in a wide range of real-world graphs including friendship networks (McPherson et al., 2001), political networks (Gerber et al., 2013; Newman, 2018), citation networks (Ciotti et al., 2016) and more. Under the homophily assumption, through the aggregation process, a node's representation is "smoothed" via its neighbors' representations, since each node is able to receive additional information from neighboring nodes, which are likely to share the same label. Several recent works (Zhu et al., 2020b;a) claim that GNNs are implicitly (or explicitly) designed with homophily in mind, are not suitable for graphs exhibiting *heterophily*, where connected nodes are prone to have different properties or labels, e.g dating networks or molecular networks (Zhu et al., 2020b). Such works accordingly design and modify new architectures and demonstrate outperformance over other GNN models on several heterophilous graphs.

**Present work.** In our work, we empirically find that the graph convolutional network (GCN) Kipf and Welling (2016), a fundamental, representative GNN model (which we focus on in this work) is actually able to outperform such heterophily-specific models on some heterophilous graphs after careful hyperparameter tuning. This motivates us to reconsider the popular notion in the literature that GNNs exhibit a homophilous inductive bias, and more specifically that strong homophily is crucial to strong GNN performance. Counter to this idea, we find that GCN model has the potential to work well for heterophilous graphs under suitable conditions. We demonstrate intuition with the following toy example: Consider the perfectly heterophilous graph (with all inter-class edges) shown in Figure 1, where the color indicates the node label. Blue-labeled and orange-labeled nodes are associated with the scalar feature $0$ and $1$, respectively. If we consider a single-layer GCN by performing an averaging feature aggregation over all neighboring nodes, it is clear that all blue nodes will have a representation of $1$, while the orange nodes will have that of $0$. Additional layers/aggregations will continue to alternate the features between the two types of nodes. Regardless of the number of layers, the two classes can still be perfectly separated. In this toy example, each blue (orange) node only connects orange (blue) nodes, and all blue (orange) nodes share similar neighborhood patterns in terms of their neighbors' label/feature distributions.

Our work elucidates this intuition and extends it to a more general case: put simply, given a (homophilous or heterophilous) graph, GCN has the potential to achieve good performance if nodes with the same label share similar neighborhood patterns. We theoretically support this argument by investigating the learned node embeddings from the GCN model. We find that homophilous graphs always satisfy such assumptions, which explains why GCN typically works well for them. On other hand, there exist both "good" and "bad" heterophily, and GCNs can actually achieve strong performance for "good" heterophily settings while they usually fail on "bad" heterophily settings. Our work characterizes these settings, and provides a new perspective and solid step towards deeper understanding for heterophilous graphs. In short:

**Our contributions.** (1) We reveal that strong homophily is not a necessary assumption for the GCN model. The GCN model can perform well over some heterophilous graphs under certain conditions. (2) We carefully characterize these conditions and provide theoretical understandings on how GCNs can achieve good SSNC performance under these conditions by investigating their embedding learning process. (3) We carefully investigate commonly used homophilous and heterophilous benchmarks and reason about GCN's performs on them utilizing our theoretical understanding.

## 2  PRELIMINARIES

Let $\mathcal{G} = \{\mathcal{V}, \mathcal{E}\}$ denote a graph, where $\mathcal{V}$ and $\mathcal{E}$ are the sets of nodes and edges, respectively. The graph connection information can also be represented as an adjacency matrix $\mathbf{A} \in \{0, 1\}^{|\mathcal{V}| \times |\mathcal{V}|}$, where $|\mathcal{V}|$ is the number of nodes in the graph. The $i, j$-th element of the adjacency matrix $\mathbf{A}[i, j]$ is equal to $1$ if and only if nodes $i$ and $j$ are adjacent to each other, otherwise $\mathbf{A}[i, j] = 0$. Each node $i$ is associated with a $l$-dimensional vector of node features $\mathbf{x}_i \in \mathbb{R}^l$; the features for all nodes can be summarized as a matrix $\mathbf{X} \in \mathbb{R}^{|\mathcal{V}| \times l}$. Furthermore, each node $i$ is associated with a label $y_i \in \mathcal{C}$, where $\mathcal{C}$ denotes the set of labels. We also denote the set of nodes with a given label $c \in \mathcal{C}$ as $\mathcal{V}_c$. We assume that labels are only given for a subset of nodes $\mathcal{V}_{label} \subset \mathcal{V}$. The goal of semi-supervised node classification (SSNC) is to learn a mapping $f : \mathcal{V} \to \mathcal{C}$ utilizing the graph $\mathcal{G}$, the node features $\mathbf{X}$ and the labels for nodes in $\mathcal{V}_{label}$.

### 2.1  HOMOPHILY IN GRAPHS

In this work, we focus on investigating performance in the context of graph homophily and heterophily properties. Homophily in graphs is typically defined based on similarity between connected node pairs, where two nodes are considered similar if they share the same node label. The homophily ratio is defined based on this intuition following Zhu et al. (2020b).

**Definition 1** (Homophily). *Given a graph $\mathcal{G} = \{\mathcal{V}, \mathcal{E}\}$ and node label vector $y$, the edge homophily ratio is defined as the fraction of edges that connect nodes with the same labels. Formally, we have:*

$$h(\mathcal{G}, \{y_i; i \in \mathcal{V}\}) = \frac{1}{|\mathcal{E}|} \sum_{(j,k) \in \mathcal{E}} \mathbb{1}(y_j = y_k), \tag{1}$$

*where $|\mathcal{E}|$ is the number of edges in the graph and $\mathbb{1}(\cdot)$ is the indicator function.*

A graph is typically considered to be highly homophilous when $h(\cdot)$ is large (typically, $0.5 \leq h(\cdot) \leq 1$), given suitable label context. On the other hand, a graph with a low edge homophily ratio is considered to be heterophilous. In future discourse, we write $h(\cdot)$ as $h$ when discussing given a fixed graph and label context.

## 2.2 GRAPH NEURAL NETWORKS

Graph neural networks learn node representations by aggregating and transforming information over the graph structure. There are different designs and architectures for the aggregation and transformation, which leads to different graph neural network models (Scarselli et al., 2008; Kipf and Welling, 2016; Hamilton et al., 2017; Veličković et al., 2017; Gilmer et al., 2017; Zhou et al., 2020).

One of the most popular and widely adopted GNN models is the graph convolutional network (GCN). A single GCN operation takes the following form $\mathbf{H}' = \mathbf{D}^{-1}\mathbf{A}\mathbf{H}\mathbf{W}$, where $\mathbf{H}$ and $\mathbf{H}'$ denote the input and output features of layer, $\mathbf{W}^{(k)} \in \mathbb{R}^{l \times l}$ is a parameter matrix to transform the features, and $\mathbf{D}$ is a diagonal matrix and $\mathbf{D}[i,i] = deg(i)$ with $deg(i)$ denoting the degree of node $i$. From a local perspective for node $i$, the process can be written as a feature averaging process $\mathbf{h}_i = \frac{1}{deg(i)}\sum_{j \in \mathcal{N}(i)} \mathbf{W}\mathbf{x}_j$, where $\mathcal{N}(i)$ denotes the neighbors of node $i$. The neighborhood $\mathcal{N}(i)$ may contain the node $i$ itself. Usually, when building GCN model upon GCN operations, nonlinear activation functions are added between consecutive GCN operations.

## 3 GRAPH CONVOLUTIONAL NETWORKS UNDER HETEROPHILY

Considerable prior literature posits that graph neural networks (such as GCN) work by assuming and exploiting homophily assumptions in the underlying graph (Maurya et al., 2021; Halcrow et al., 2020; Wu et al., 2018). To this end, researchers have determined that such models are considered to be ill-suited for heterophilous graphs, where the homophily ratio is low (Zhu et al., 2020b;a; Chien et al., 2021). To deal with this limitation, researchers proposed several methods including H2GNN (Zhu et al., 2020b), CPGNN (Zhu et al., 2020a) and GPRGNN (Chien et al., 2021), which are explicitly designed to handle heterophilous graphs via architectural choices (e.g. adding skip-connections, carefully choosing aggregators, etc.)

In this section, we revisit the claim that GCNs have fundamental homophily assumptions and are not suited for heterophilous graphs. To this end, we first observe empirically that the GCN model achieves fairly good performance on some of the commonly used heterophilous graphs; specifically, we present SSNC performance on two commonly used heterophilous graph datasets, Chameleon and Squirrel in Table 1 (see Appendix D for further details about the datasets and models). Both Chameleon and Squirrel are highly

Table 1: SSNC accuracy on two heterophilous datasets.

| Method | Chameleon ($h = 0.23$) | Squirrel ($h = 0.22$) |
|---|---|---|
| GCN | **67.96** ± 1.82 | **54.47** ± 1.17 |
| H2GCN-1 | 57.11 ± 1.58 | 36.42 ± 1.89 |
| H2GCN-2 | 59.39 ± 1.98 | 37.90 ± 2.02 |
| CPGNN-MLP | 54.53 ± 2.37 | 29.13 ± 1.57 |
| CPGNN-Cheby | 65.17 ± 3.17 | 29.25 ± 4.17 |
| GPRGNN | 66.31 ± 2.05 | 50.56 ± 1.51 |
| MLP | 48.11 ± 2.23 | 31.68 ± 1.90 |

heterophilous ($h \approx 0.2$). We find that with some hyperparameter tuning, GCN can *outperform alternative methods uniquely designed to operate on some certain heterophilous graphs*. This observation suggests that GCN does not always "underperform" on heterophilous graphs, and it leads us to reconsider the prevalent assumption in literature. Hence, we next examine how GCNs learn representations, and how this information is used in downstream SSNC tasks.

### 3.1 WHEN DOES GCN LEARN SIMILAR EMBEDDINGS FOR NODES WITH THE SAME LABEL?

GCN is considered to be unable to tackle heterophilous graphs due to its feature averaging process (Zhu et al., 2020b; Chien et al., 2021). Namely, a node's newly aggregated features are considered "corrupted" by those neighbors that do not share the same label, leading to the intuition that GCN embeddings are noisy and un-ideal for SSNC. However, we find that crucially, for some heterophilous graphs, the features of nodes with the same label are "corrupted in the same way." Hence, the obtained embeddings still contain informative characteristics and thus facilitate SSNC. We next illustrate when GCN learns similar embeddings for nodes with the same label, beginning with a toy example and generalizing to more practical cases.

GCNs have been shown to be able to capture the local graph topological and structural information (Xu et al., 2019; Morris et al., 2019). Specifically, the aggregation step in the GCN model is able to capture and discriminate neighborhood distribution information, e.g. the mean of the neighborhood features (Xu et al., 2019). Let us con-

Figure 2: Two nodes share the same neighborhood distribution; GCN learns equivalent embeddings for $a$ and $b$.

sider the two nodes $a$ and $b$ shown in Figure 2, where we use color to indicate the label of each node. If we further assume that all nodes sharing the same label are associated with exactly the same features, then clearly, after 1-step aggregation, the GCN operation will output exactly the same embedding for nodes $a$ and $b$. Accordingly, Xu et al. (2019) reasons that the GCN model lacks expressiveness due to its inability to differentiate the two nodes in the embedding space. However, in

the SSNC task, mapping $a$ and $b$ to the same location in the embedding space is explicitly desirable. Intuitively, if all nodes with the same label are mapped to the same embedding and embeddings for different labels are distinct, SSNC is effortless (Zhao et al., 2020).

Such assumptions are hard to meet in practice. Thus, to consider a more practical scenario, we assume that both features and neighborhood patterns for nodes with a certain label are sampled from some fixed distributions. Under these conditions, same-label nodes may not share fixed embeddings, but we can aim to characterize their closeness. Intuitively, if the learned embeddings for same-label nodes are close and embeddings for other-label nodes are far, we expect strong SSNC performance to be good, given class separability (low intra-class variance and high inter-class variance) (Fisher, 1936). We prove that, for graphs meeting suitable conditions the distance between GCN-learned embeddings of any same-label node pair is bounded by a small quantity with high probability.

**Assumptions on Graphs.** We consider a graph $\mathcal{G}$, where each node $i$ has features $\mathbf{x}_i \in \mathbb{R}^l$ and label $y_i$. We assume that (1) The features of node $i$ are sampled from feature distribution $\mathcal{F}_{y_i}$, i.e, $\mathbf{x}_i \sim \mathcal{F}_{y_i}$, with $\mu(\mathcal{F}_{y_i})$ denoting its mean; (2) Dimensions of $\mathbf{x}_i$ are independent to each other; (3) The features in $\mathbf{X}$ are bounded by a positive scalar $B$, i.e, $\max_{i,j} |\mathbf{X}[i,j]| \leq B$; (4) For node $i$, its neighbor's labels are independently sampled from neighbor distribution $\mathcal{D}_{y_i}$. The sampling is repeated for $deg(i)$ times to sample the labels for $deg(i)$ neighbors.

We denote a graph following these assumptions (1)-(4) as $\mathcal{G} = \{\mathcal{V}, \mathcal{E}, \{\mathcal{F}_c, c \in \mathcal{C}\}, \{\mathcal{D}_c, c \in \mathcal{C}\}\}$. Note that we use the subscripts in $\mathcal{F}_{y_i}$ and $\mathcal{D}_{y_i}$ to indicate that these two distributions are shared by all nodes with the same label as node $i$. Next, we analyze the embeddings obtained after a GCN operation. Following previous works (Li et al., 2018; Chen et al., 2020; Baranwal et al., 2021), we drop the non-linearity in the analysis.

**Theorem 1.** *Consider a graph $\mathcal{G} = \{\mathcal{V}, \mathcal{E}, \{\mathcal{F}_c, c \in \mathcal{C}\}, \{\mathcal{D}_c, c \in \mathcal{C}\}\}$, which follows Assumptions (1)-(4). For any node $i \in \mathcal{V}$, the expectation of the pre-activation output of a single GCN operation is given by*

$$\mathbb{E}[\mathbf{h}_i] = \mathbf{W} \left( \mathbb{E}_{c \sim \mathcal{D}_{y_i}, \mathbf{x} \sim \mathcal{F}_c}[\mathbf{x}] \right). \tag{2}$$

*and for any $t > 0$, the probability that the distance between the observation $\mathbf{h}_i$ and its expectation is larger than $t$ is bounded by*

$$\mathbb{P}\left(\|\mathbf{h}_i - \mathbb{E}[\mathbf{h}_i]\|_2 \geq t\right) \leq 2 \cdot l \cdot \exp\left(-\frac{deg(i)t^2}{2\rho^2(\mathbf{W})B^2 l}\right), \tag{3}$$

*where $l$ denotes the feature dimensionality and $\rho(\mathbf{W})$ denotes the largest singular value of $\mathbf{W}$.*

The detailed proof can be found in Appendix A. Theorem 1 demonstrates two key ideas. First, in expectation, all nodes with the same label have the same embedding (Eq. (2)). Second, the distance between the output embedding of a node and its expectation is small with a high probability. Specifically, this probability is related to the node degree and higher degree nodes have higher probability to be close to the expectation. Together, these results show that the GCN model is able to map nodes with the same label to an area centered around the expectation in the embedding space under given assumptions. Then, the downstream classifier in the GCN model is able to assign these nodes to the same class with high probability. To ensure that the classifier achieves strong performance, the centers (or the expectations) of different classes must be distant from each other; if we assume that $\mu(\mathcal{F}_{y_i})$ are distinct from each other (as is common), then the neighbor distributions $\{\mathcal{D}_c, c \in \mathcal{C}\}$ must be distinguishable to ensure good SSNC performance. Based on these understandings and discussions, we have the following key (informal) observations on GCN's performance for graphs with homophily and heterophily.

**Observation 1** (GCN under Homophily). *In homophilous graphs, the neighborhood distribution of nodes with the same label (w.l.o.g $c$) can be approximately regarded as a highly skewed discrete $\mathcal{D}_c$, with most of the mass concentrated on the category $c$. Thus, different labels clearly have distinct distributions. Hence, the GCN model typically in SSNC on such graph, with high degree nodes benefiting more, which is consistent with previous work (Tang et al., 2020b).*

**Observation 2** (GCN under Heterophily). *In heterophilous graphs, if the neighborhood distribution of nodes with the same label (w.l.o.g. $c$) is (approximately) sampled from a fixed distribution $\mathcal{D}_c$, and different labels have distinguishable distributions, then GCN can excel at SSNC, especially when node degrees are large. Otherwise, GCNs may fail for heterophilous graphs.*

Notably, our findings illustrate that disruptions of certain conditions inhibit GCN performance on heterophilous graphs, but heterophily is *not a sufficient condition for poor GCN performance*. GCNs are able to achieve reasonable performance for both homophilous and heterophilous graphs if they follow certain assumptions as discussed in the two observations. In Section 3.2, we theoretically

demonstrate these observations for graphs sampled from the Contextual Stochastic Block Model (CSBM) (Deshpande et al., 2018) with two classes, whose distinguishablilty of neighborhood distributions can be explicitly characterized. Furthermore, in Section 3.3, we empirically demonstrate these observations on graphs with multiple classes. We note that although our derivations are for GCN, a similar line of analysis can be used for more general message-passing neural networks.

## 3.2 ANALYSIS BASED ON CSBM MODEL WITH TWO CLASSES

**The CSBM model.** To clearly control assumptions, we study the contextual stochastic block model(CSBM), a generative model for random graphs; such models have been previously adopted for benchmarking graph clustering (Fortunato and Hric, 2016) and GNNs (Tsitsulin et al., 2021). Specifically, we consider a CSBM model consisting of two classes $c_1$ and $c_2$. In this case, the nodes in the generated graphs consist of two disjoint sets $\mathcal{C}_1$ and $\mathcal{C}_2$ corresponding to the two classes, respectively. Edges are generated according to an intra-class probability $p$ and an inter-class probability $q$. Specifically, any two nodes in the graph, are connected by an edge with probability $p$, if they are from the same class, otherwise, the probability is $q$. For each node $i$, its initial features $\mathbf{x}_i \in \mathbb{R}^l$ are sampled from a Gaussian distribution $\mathbf{x}_i \sim N(\boldsymbol{\mu}, \mathbf{I})$, where $\boldsymbol{\mu} = \boldsymbol{\mu}_k \in \mathbb{R}^l$ for $i \in \mathcal{C}_k$ with $k \in \{1, 2\}$ and $\boldsymbol{\mu}_1 \neq \boldsymbol{\mu}_2$. We denote a graph generated from such an CSBM model as $\mathcal{G} \sim \text{CSBM}(\boldsymbol{\mu}_1, \boldsymbol{\mu}_2, p, q)$. We denote the features for node $i$ obtained after a GCN operation as $\mathbf{h}_i$.

**Linear separability under GCN.** To better evaluate the effectiveness of GCN operation, we study the linear classifiers with the largest margin based on $\{\mathbf{x}_i, i \in \mathcal{V}\}$ and $\{\mathbf{h}_i, i \in \mathcal{V}\}$ and compare their performance. Since the analysis is based on linear classifiers, we do not consider the linear transformation in the GCN operation as it can be absorbed in the linear model, i.e, we only consider the process $\mathbf{h}_i = \frac{1}{deg(i)} \sum_{j \in \mathcal{N}(i)} \mathbf{x}_j$. For a graph $\mathcal{G} \sim \text{CSBM}(\boldsymbol{\mu}_1, \boldsymbol{\mu}_2, p, q)$, we can approximately regard that for each node $i$, its neighbor's labels are independently sampled from a neighborhood distribution $\mathcal{D}_{y_i}$, where $y_i$ denotes the label of node $i$. Specifically, the neighborhood distributions corresponding to $c_1$ and $c_2$ are $\mathcal{D}_{c_1} = [\frac{p}{p+q}, \frac{q}{p+q}]$ and $\mathcal{D}_{c_2} = [\frac{q}{p+q}, \frac{p}{p+q}]$, respectively.

Based on the neighborhood distributions, the features obtained from GCN operation follow Gaussian distributions:

$$\mathbf{h}_i \sim N\left(\frac{p\boldsymbol{\mu}_1 + q\boldsymbol{\mu}_2}{p + q}, \frac{\mathbf{I}}{\sqrt{deg(i)}}\right), \text{for } i \in \mathcal{C}_1; \text{ and } \mathbf{h}_i \sim N\left(\frac{q\boldsymbol{\mu}_1 + p\boldsymbol{\mu}_2}{p + q}, \frac{\mathbf{I}}{\sqrt{deg(i)}}\right), \text{for } i \in \mathcal{C}_2. \quad (4)$$

Based on the properties of Gaussian distributions, it is easy to see that Theorem 1 holds. We denote the expectation of the original features for nodes in the two classes as $\mathbb{E}_{c_1}[\mathbf{x}_i]$ and $\mathbb{E}_{c_2}[\mathbf{x}_i]$. Similarly, we denote the expectation of the features obtained from GCN operation as $\mathbb{E}_{c_1}[\mathbf{h}_i]$ and $\mathbb{E}_{c_2}[\mathbf{h}_i]$. The following proposition describes their relations.

**Proposition 1.** $(\mathbb{E}_{c_1}[\mathbf{x}_i], \mathbb{E}_{c_2}[\mathbf{x}_i])$ and $(\mathbb{E}_{c_1}[\mathbf{h}_i], \mathbb{E}_{c_2}[\mathbf{h}_i])$ share the same middle point. $\mathbb{E}_{c_1}[\mathbf{x}_i] - \mathbb{E}_{c_2}[\mathbf{x}_i]$ and $\mathbb{E}_{c_1}[\mathbf{h}_i] - \mathbb{E}_{c_2}[\mathbf{h}_i]$ share the same direction. Specifically, the middle point $\mathbf{m}$ and the shared direction $\mathbf{w}$ are as follows: $\mathbf{m} = (\boldsymbol{\mu}_1 + \boldsymbol{\mu}_2)/2$, and $\mathbf{w} = (\boldsymbol{\mu}_1 - \boldsymbol{\mu}_2)/\|\boldsymbol{\mu}_1 - \boldsymbol{\mu}_2\|_2$.

This proposition follows from direct calculations. Given that the feature distributions of these two classes are systematic to each other (for both $\mathbf{x}_i$ and $\mathbf{h}_i$), the hyperplane that is orthogonal to $\mathbf{w}$ and goes through $\mathbf{m}$ defines the decision boundary of the optimal linear classifier for both types of features. We denote this decision boundary as $\mathcal{P} = \{\mathbf{x} | \mathbf{w}^\top \mathbf{x} - \mathbf{w}^\top (\boldsymbol{\mu}_1 + \boldsymbol{\mu}_2)/2\}$.

Next, to evaluate how GCN operation affects the classification performance, we compare the probability that this linear classifier misclassifies a certain node based on the features before and after the GCN operation. We summarize the results in the following theorem.

**Theorem 2.** Consider a graph $\mathcal{G} \sim CSBM(\boldsymbol{\mu}_1, \boldsymbol{\mu}_2, p, q)$. For any node $i$ in this graph, the linear classifier defined by the decision boundary $\mathcal{P}$ has a lower probability to misclassify $\mathbf{h}_i$ than $\mathbf{x}_i$ when $deg(i) > (p + q)^2/(p - q)^2$.

The detailed proof can be found in Appendix B. Note that the Euclidean distance between the two discrete neighborhood distributions $\mathcal{D}_{c_0}$ and $\mathcal{D}_{c_1}$ is $\sqrt{2}\frac{|p-q|}{(p+q)}$. Hence, Theorem 2 demonstrates that the node degree $deg(i)$ and the distinguishability (measured by the Euclidean distance) of the neighborhood distributions both affect GCN's performance. Specifically, we can make the following conclusions: (1) When $p$ and $q$ are fixed, the GCN operation is more likely to improve the linear separability of the high-degree nodes than low-degree nodes, which is consistent with observations in (Tang et al., 2020b). (2) The more distinguishable the neighborhood distributions are (or the larger the Euclidean distance is), the more nodes can be benefited from the GCN operation. For

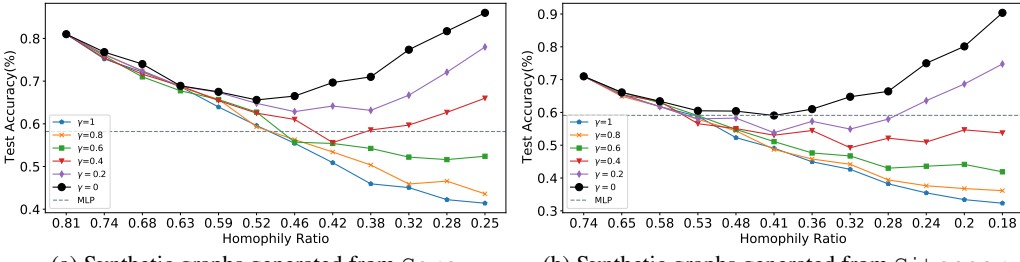

(a) Synthetic graphs generated from `Cora`          (b) Synthetic graphs generated from `Citeseer`

Figure 3: SSNC accuracy of GCN on synthetic graphs with various homophily ratios.

example, when $p = 9q$ or $9p = q$, $(p + q)^2/(p - q)^2 \approx 1.23$, thus nodes with degree larger than $1$ can benefit from the GCN operation. These two cases correspond to extremely homophily ($h = 0.9$) and extremely heterophily ($h = 0.1$), respectively. However, GCN model behaves similarly on these two cases and is able to improve the performance for most nodes. This clearly demonstrates that heterophily is *not a sufficient condition* for poor GCN performance. Likewise, when $p \approx q$, the two neighborhood distributions are hardly distinguishable, and only nodes with extremely large degrees can benefit. In the extreme case, where $p = q$, the GCN operation cannot help any nodes at all. Note that Theorem 2 and the followed analysis can be extended to a multi-class CSBM scenario as well – see Appendix F for an intuitive explanation and proof sketch.

### 3.3 EMPIRICAL INVESTIGATIONS ON GRAPHS WITH MULTIPLE CLASSES

We conduct experiments to substantiate our claims in Observations 1 and 2 in graphs with multiple classes. We evaluate how SSNC performance changes as we make a homophilous graph more and more heterophilous under two settings: (1) different labels have distinct distributions, and (2) different labels' distributions are muddled.

#### 3.3.1 TARGETED HETEROPHILOUS EDGE ADDITION

**Graph generation strategy.** We start with common, real-world benchmark graphs, and modify their topology by adding synthetic, cross-label edges that connect nodes with different labels. Following our discussion in Observation 2, we construct synthetic graphs that have similar neighborhood distributions for same-label nodes. Specifically, given a real-world graph $\mathcal{G}$, we first define a discrete neighborhood target distribution $\mathcal{D}_c$ for each label $c \in \mathcal{C}$. We then follow these target distributions to add cross-label edges. The process of generating new graphs by adding edges to $\mathcal{G}$ is shown in Algorithm 1. Specifically, we add a total $K$ edges to the given graph $\mathcal{G}$: to add each edge, we first uniformly sample a node $i$ from $\mathcal{V}$ with label $y_i$, then we sample a label $c$ from $\mathcal{C}$ according to $\mathcal{D}_{y_i}$, and finally, we uniformly sample a node $j$ from $\mathcal{V}_c$ and add the edge $(i, j)$ to the graph. We generate synthetic graphs based on several real-world graphs. We present the results based on `Cora` and `Citeseer` (Sen et al., 2008). The results for other datasets can be found in Appendix C. Both `Cora` and `Citeseer` exhibit strong homophily. For both datasets, we fix $\mathcal{D}_c$ for all labels. Although many suitable $\mathcal{D}_c$ could be specified in line with Observation 2, we fix one set for illustration and brevity. For both datasets, we vary $K$ over 11 values and thus generate 11 graphs. Notably, as $K$ increases, the homophily $h$ decreases. More detailed information about the $\{\mathcal{D}_c, c \in \mathcal{C}\}$ and $K$ for both datasets is included in Appendix C.

---

**Alg. 1:** Hetero. Edge Addition

**input** : $\mathcal{G} = \{\mathcal{V}, \mathcal{E}\}, K, \{\mathcal{D}_c\}_{c=0}^{|\mathcal{C}|-1}$
  and $\{\mathcal{V}_c\}_{c=0}^{|\mathcal{C}|-1}$
**output:** $\mathcal{G}' = \{\mathcal{V}, \mathcal{E}'\}$
Initialize $\mathcal{G}' = \{\mathcal{V}, \mathcal{E}\}, k = 1$ ;
**while** $1 \le k \le K$ **do**
  Sample node $i \sim \text{Uniform}(\mathcal{V})$;
  Obtain the label, $y_i$ of node $i$;
  Sample a label $c \sim \mathcal{D}_{y_i}$;
  Sample node $j \sim \text{Uniform}(\mathcal{V}_c)$;
  Update edge set $\mathcal{E}' = \mathcal{E}' \cup \{(i, j)\}$;
  $k \leftarrow k + 1$;

**return** $\mathcal{G}' = \{\mathcal{V}, \mathcal{E}'\}$

---

**Observed results.** Figure 3(a-b) show SSNC results (accuracy) on graphs generated based on `Cora` and `Citeseer`, respectively. The black line in both figures shows results for the presented setting (we introduce $\gamma$ in next subsection). Without loss of generality, we use `Cora` (a) to discuss our findings, since observations are similar over these datasets. Each point on the black line in Figure 3(a) represents the performance of GCN model on a certain generated graph and the corresponding value in $x$-axis denotes the homophily ratio of this graph. The point with homophily ratio $h = 0.81$ denotes the original `Cora` graph, i.e, $K = 0$. We observe that as $K$ increases, $h$ decreases, and while the classification performance first decreases, it *eventually begins to increase*, showing a $V$-shape pattern. For instance, when $h = 0.25$ (a rather heterophilous graph), the GCN model achieves an impressive $86\%$ accuracy, even higher than that achieved on the original `Cora` graph. We note that

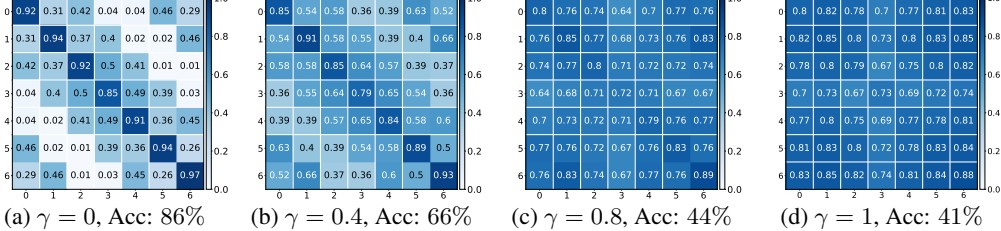

Figure 4: Cross-class neighborhood similarity on synthetic graphs generated from Cora; all graphs have $h = 0.25$, but with varying neighborhood distributions as per the noise parameter $\gamma$.

performance continues to increase as $K$ increases further to the right (we censor due to space limitations; see Appendix C for details). This clearly demonstrates that the GCN model can work well on heterophilous graphs under certain conditions. Intuitively, the $V$-shape arises due to a "phase transition", where the initial topology is overridden by added edges according to the associated $\mathcal{D}_c$ target neighbor distributions. In the original graph, the homophily ratio is quite high ($h = 0.81$), and classification behavior is akin to that discussed in Observation 1, where same-label nodes have similar neighborhood patterns. As we add edges to the graph, the originally evident neighborhood patterns are perturbed by added edges and gradually become less informative, which leads to the performance decrease in the decreasing segment of the $V$-shape in Figure 3(a). Then, as we keep adding more edges, the neighborhood pattern gradually approaches $\mathcal{D}_c$ for all $c$, corresponding to the increasing segment of the $V$-shape.

### 3.3.2 INTRODUCING NOISE TO NEIGHBORHOOD DISTRIBUTIONS

**Graph generation strategy.** In Section 3.3.1, we showed that the GCN model can achieve reasonable performance on heterophilous graphs constructed following distinct, pre-defined neighborhood patterns. As per Observation 2, our theoretical understanding suggests that performance should degrade under heterophily if the distributions of different labels get more and more indistinguishable. Hence, we next demonstrate this empirically by introducing controllable noise levels into our edge addition strategy. We adopt a strategy similar to that described in Algorithm 1, but with the key difference being that we introduce an additional parameter $\gamma$, which controls the probability that we add cross-label edges *randomly* rather than following the pre-defined distributions. A detailed description of this approach is demonstrated in Algorithm 2 in Appendix C. For nodes of a given class $c$ (w.l.o.g), compared to the edges added according to $\mathcal{D}_c$, the randomly added edges can be regarded as noise. Specifically, by increasing the noise parameter $\gamma$, we increase the similarity between $\mathcal{D}_c, \mathcal{D}_{c'}$ for any pair of labels $c, c'$. If $\gamma = 1$, then all neighborhood distributions will be indistinguishable (they will all be approximately $\mathsf{Uniform}(|\mathcal{C}|)$). By fixing $K$ and varying $\gamma$, we can generate graph variants with the same homophily ratio but different similarities between $\mathcal{D}_c$ and $\mathcal{D}_{c'}$. As in Section 3.3.1, we create graphs by adding edges at various $K$, but also vary $\gamma \in [0, 1]$ in increments of $0.2$ on both Cora and Citeseer.

**Observed results.** We report the SSNC performance on these graphs in Figure 3. Firstly, we observe that noise affects the performance significantly when the homophily ratio is low. For example, observing Figure 3(a) vertically at homophily ratio $h = 0.25$, higher $\gamma$ clearly results in worse performance. This indicates that not only the fixed-ness of the neighborhood distributions, but their similarities are important for the SSNC task (aligned with Observation 2. It also indicates that there are "good" and "bad" kinds of heterophily. On the other hand, high $\gamma$ does not too-negatively impact when $K$ is small, since noise is minimal and the original graph topology is yet largely homophilous. At this stage, both "good" (fixed and disparate patterns) and "bad" (randomly added edges) heterophilous edges introduce noise to the dominant homophilous patterns. When the noise level $\gamma$ is not too large, we can still observe the $V$-shape: e.g. $\gamma = 0.4$ in Figure 3(a) and $\gamma = 0.2$ in Figure 3 (b); this is because the designed pattern is not totally dominated by the noise. However, when $\gamma$ is too high, adding edges will constantly decrease the performance, as nodes of different classes have indistinguishably similar neighborhoods.

To further demonstrate how $\gamma$ affects the neighborhood distributions in the generated graph, we examine the cross-class neighborhood similarity, which we define as follows:

**Definition 2** (Cross-Class Neighborhood Similarity (CCNS)). *Given graph $\mathcal{G}$ and labels $\mathbf{y}$ for all nodes, the CCNS between classes $c, c' \in \mathcal{C}$ is $s(c, c') = \frac{1}{|\mathcal{V}_c||\mathcal{V}_{c'}|} \sum_{i \in \mathcal{V}_c, j \in \mathcal{V}_{c'}} \cos{(d(i), d(j))}$ where*

$\mathcal{V}_c$ *indicates the set of nodes in class $c$ and $d(i)$ denotes the empirical histogram (over $|\mathcal{C}|$ classes) of node $i$'s neighbors' labels, and the function $\cos(\cdot, \cdot)$ measures the cosine similarity.*

When $c = c'$, $s(c, c')$ calculates the intra-class similarity, otherwise, it calculates the inter-class similarity from a neighborhood label distribution perspective. Intuitively, if nodes with the same label share the same neighborhood distributions, the intra-class similarity should be high. Likewise, to ensure that the neighborhood patterns for nodes with different labels are distinguishable, the inter-class similarity should be low. To illustrate how various $\gamma$ values affect the neighborhood patterns, we illustrate the intra-class and inter-class similarities in Figure 4 for $\gamma = 0, 0.4, 0.8, 1$ on graphs generated from `Cora` with homophily ratio $h = 0.25$. The diagonal cells in each heatmap indicate the intra-class similarity while off-diagonal cells indicate inter-class similarity. Clearly, when $\gamma$ is small, the intra-class similarity is high while the inter-class similarity is low, which demonstrates the existence of strongly discriminative neighborhood patterns in the graph. As $\gamma$ increases, the intra-class and inter-class similarity get closer, becoming more and more indistinguishable, leading to bad performance due to indistinguishable distributions as referenced in Observation 2.

## 4    REVISITING GCN'S PERFORMANCE ON REAL-WORLD GRAPHS

In this section, we first give more details on the experiments we run to compare GCN and MLP. We next investigate why the GCN model does or does not work well on certain datasets utilizing the understanding developed in earlier sections.

### 4.1    QUANTITATIVE ANALYSIS

Following previous work (Pei et al., 2020; Zhu et al., 2020b), we evaluate the performance of the GCN model on several real-world graphs with different levels of homophily. We include the citation networks `Cora`, `Citeseer` and `Pubmed` (Kipf and Welling, 2016), which are highly homophilous. We also adopt several heterophilous benchmark datasets including `Chameleon`, `Squirrel`, `Actor`, `Cornell`, `Wisconsin` and `Texas` (Rozemberczki et al., 2021; Pei et al., 2020). Appendix D.1 gives descriptions and summary statistics of these datasets. For all datasets, we follow the experimental setting provided in (Pei et al., 2020), which consists of 10 random splits with proportions $48/32/20\%$ corresponding to training/validation/test for each graph. For each split, we use 10 random seeds, and report the average performance and standard deviation across 100 runs. We compare the GCN model with the MLP model, which does not utilize the graph structure. With this comparison, we aim to check whether the GCN model always fails for for heterophilous graphs (perform even worse than MLP). We also compare GCN with state-of-the-art methods and their descriptions and performance are included in the Appendix D. The node classification performance (accuracy) of these models is reported in Table 2. Notably, GCN achieves better performance than MLP on graphs with high homophily (`Cora`, `Citeseer`, and `Pubmed`), as expected. For the heterophilous graphs, the results are comparatively mixed. The GCN model outperforms MLP on `Squirrel` and `Chameleon` (it even outperforms methods specifically designed for heterophilous graphs as shown in Appendix D.5.), while underperforming on the other datasets (`Actor`, `Cornell`, `Wisconsin`, and `Texas`). In the next section, we provide explanations for GCN's distinct behaviors on these graphs based on the understanding developed in earlier sections.

### 4.2    QUALITATIVE ANALYSIS

Our work so far illustrates that the popular notion of GCNs not being suitable for heterophily, or homophily being a mandate for good GCN performance is not accurate. In this subsection, we aim to use the understanding we developed in Section 3 to explain why GCN does (not) work well on real-world graphs. As in Section 3.3.2, we inspect cross-class neighborhood similarity (Definition 2) for each dataset; due to the space limit, we only include representative ones here (`Cora`, `Chameleon`, `Actor` and `Cornell`; see Figure 5). Heatmaps for the other datasets can be found in Appendix E. From Figure 5(a), it is clear that the intra-class similarity is much higher than the inter-similarity ones, hence `Cora` contains distinct neighborhood patterns, consistent with Observation 1. In Figure 5(b), we can observe that in `Chameleon`, intra-class similarity is generally higher than inter-class similarity, though not as strong as in Figure 5(a). Additionally, there is an apparent gap between labels $0, 1$ and $2, 3, 4$, which contributes to separating nodes of the former 2 from the latter 3 classes, but potentially increasing misclassification within each of the two groupings. These observations also help substantiate why GCN can achieve reasonable performance (much higher than MLP) on `Chameleon`. The GCN model underperforms MLP in `Actor` and we suspect that the graph does not provide useful information. The heatmap for `Actor` in Figure 5 shows that the intra-class and inter-class similarities are almost equivalent, making the neighborhood

Table 2: Node classification performance (accuracy) on homophilous and heterophilous graphs.

| | Cora | Citeseer | Pubmed | Chameleon | Squirrel | Actor | Cornell | Wisconsin | Texas |
|---|---|---|---|---|---|---|---|---|---|
| GCN | $87.12 \pm 1.38$ | $76.50 \pm 1.61$ | $88.52 \pm 0.41$ | $67.96 \pm 1.82$ | $54.47 \pm 1.17$ | $30.31 \pm 0.98$ | $59.35 \pm 4.19$ | $61.76 \pm 6.15$ | $63.81 \pm 5.27$ |
| MLP | $75.04 \pm 1.97$ | $72.40 \pm 1.97$ | $87.84 \pm 0.30$ | $48.11 \pm 2.23$ | $31.68 \pm 1.90$ | $36.17 \pm 1.09$ | $84.86 \pm 6.04$ | $86.29 \pm 4.50$ | $83.30 \pm 4.54$ |

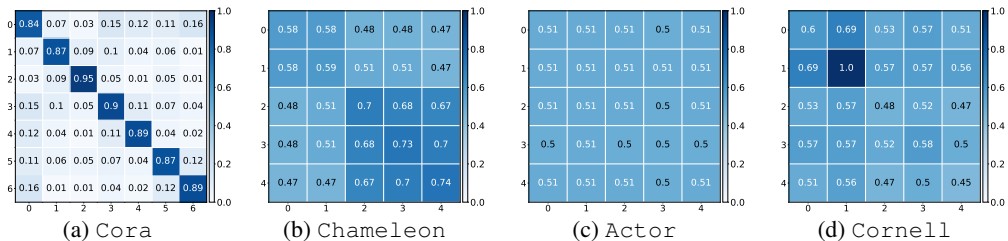

(a) Cora   (b) Chameleon   (c) Actor   (d) Cornell

Figure 5: Cross-class neighborhood similarity on homophilous graphs and heterophilous graphs.

distributions for different classes hard to distinguish and leading to bad GCN performance. Similar observations are made for Cornell. Note that Cornell only consists of 183 nodes and 280 edges, hence, the similarities shown in Figure 5 are impacted significantly (e.g. there is a single node with label 1, leading to perfect intra-class similarity for label 1).

## 5 RELATED WORK

Graph neural networks (GNNs) are powerful models for graph representation learning. They have been widely adopted to tackle numerous applications from various domains (Kipf and Welling, 2016; Fan et al., 2019; Bastings et al., 2017; Shi et al., 2019). (Scarselli et al., 2008) proposed the first GNN model, to tackle both node and graph level tasks. Subsequently, Bruna et al. (2013) and Defferrard et al. (2016) generalized convolutional neural networks to graphs from the graph spectral perspective. Kipf and Welling (2016) simplified the spectral GNN model and proposed graph convolutional networks (GCNs). Since then, numerous GNN variants, which follow specific forms of feature transformation (linear layers) and aggregation have been proposed (Veličković et al., 2017; Hamilton et al., 2017; Gilmer et al., 2017; Klicpera et al., 2019). The aggregation process can be usually understood as feature smoothing (Li et al., 2018; Ma et al., 2020; Jia and Benson, 2021; Zhu et al., 2021). Hence, several recent works claim (Zhu et al., 2020b;a; Chien et al., 2021), assume (Halcrow et al., 2020; Wu et al., 2018; Zhao et al., 2020) or remark upon (Abu-El-Haija et al., 2019; Maurya et al., 2021; Hou et al., 2020) GNN models homophily-reliance or unsuitability in capturing heterophily. Several recent works specifically develop GNN models choices to tackle heterophilous graphs by carefully designing or modifying model architectures such as Geom-GCN (Pei et al., 2020), H2GCN (Zhu et al., 2020b), GPR-GNN (Chien et al., 2021), and CPGNN (Zhu et al., 2020a). Some other works aim to modify/construct graphs to be more homophilous (Suresh et al., 2021). There is concurrent work (Luan et al., 2021) also pointing out that GCN can potentially achieve strong performance on heterophilous graphs. A major focus of this paper is to propose a new model to handle heterophilous graphs. However, our work aims to empirically and theoretically understand whether homophily is a necessity for GNNs and how GNNs work for heterophilous graphs.

## 6 CONCLUSION

It is widely believed that GNN models inherently assume strong homophily and hence fail to generalize to graphs with heterophily. In this paper, we revisit this popular notion and show it is not quite accurate. We investigate one representative model, GCN, and show empirically that it can achieve good performance on some heterophilous graphs under certain conditions. We analyze theoretically the conditions required for GCNs to learn similar embeddings for same-label nodes, facilitating the SSNC task; put simply, when nodes with the same label share similar neighborhood patterns, and different classes have distinguishable patterns, GCN can achieve strong class separation, regardless of homophily or heterophily properties. Empirical analysis supports our theoretical findings. Finally, we revisit several existing homophilous and heterophilous SSNC benchmark graphs, and investigate GCN's empirical performance in light of our understanding. Note that while there exist graphs with "good heterophily", "bad heterophily" still poses challenges to GNN models, which calls for dedicated efforts. We discuss the limitation of the current work in Appendix I.

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

# A   PROOF OF THEOREM 1

To prove Theorem 1, we first introduce the celebrated Hoeffding inequality below.

**Lemma 1** (Hoeffding's Inequality). *Let $Z_1, \ldots, Z_n$ be independent bounded random variables with $Z_i \in [a, b]$ for all $i$, where $-\infty < a \le b < \infty$. Then*

$$\mathbb{P}\left(\frac{1}{n}\sum_{i=1}^{n}(Z_i - \mathbb{E}[Z_i]) \ge t\right) \le \exp\left(-\frac{2nt^2}{(b-a)^2}\right)$$

*and*

$$\mathbb{P}\left(\frac{1}{n}\sum_{i=1}^{n}(Z_i - \mathbb{E}[Z_i]) \le -t\right) \le \exp\left(-\frac{2nt^2}{(b-a)^2}\right)$$

*for all $t \ge 0$.*

**Theorem 1.** *Consider a graph $\mathcal{G} = \{\mathcal{V}, \mathcal{E}, \{\mathcal{F}_c, c \in \mathcal{C}\}, \{\mathcal{D}_c, c \in \mathcal{C}\}\}$, which follows Assumptions (1)-(4). For any node $i \in \mathcal{V}$, the expectation of the pre-activation output of a single GCN operation is given by*

$$\mathbb{E}[\mathbf{h}_i] = \mathbf{W}\left(\mathbb{E}_{c \sim \mathcal{D}_{y_i}, \mathbf{x} \sim \mathcal{F}_c}[\mathbf{x}]\right). \tag{5}$$

*and for any $t > 0$, the probability that the distance between the observation $\mathbf{h}_i$ and its expectation is larger than $t$ is bounded by*

$$\mathbb{P}\left(\|\mathbf{h}_i - \mathbb{E}[\mathbf{h}_i]\|_2 \ge t\right) \le 2 \cdot l \cdot \exp\left(-\frac{deg(i)t^2}{2\rho^2(\mathbf{W})B^2 l}\right), \tag{6}$$

*where $l$ is the feature dimensionality and $\rho(\mathbf{W})$ denotes the largest singular value of $\mathbf{W}$.*

*Proof.* The expectation of $\mathbf{h}_i$ can be derived as follows.

$$\mathbb{E}[\mathbf{h}_i] = \mathbb{E}\left[\sum_{j \in \mathcal{N}(i)} \frac{1}{deg(i)} \mathbf{W}\mathbf{x}_j\right]$$

$$= \frac{1}{deg(i)} \sum_{j \in \mathcal{N}(i)} \mathbf{W}\mathbb{E}_{c \sim \mathcal{D}_{y_i}, \mathbf{x} \sim \mathcal{F}_c}[\mathbf{x}]$$

$$= \mathbf{W}\left(\mathbb{E}_{c \sim \mathcal{D}_{y_i}, \mathbf{x} \sim \mathcal{F}_c}[\mathbf{x}]\right).$$

We utilize Hoeffding's Inequality to prove the bound in Eq. (6). Let $\mathbf{x}_i[k], k = 1, \ldots, l$ denote the $i$-th element of $\mathbf{x}$. Then, for any dimension $k$, $\{\mathbf{x}_j[k], j \in \mathcal{N}(i)\}$ is a set of independent bounded random variables. Hence, directly applying Hoeffding's inequality, for any $t_1 \ge 0$, we have the following bound:

$$\mathbb{P}\left(\left|\sum_{j \in \mathcal{N}(i)} (\mathbf{x}_j[k] - \mathbb{E}[\mathbf{x}_j[k]])\right| \ge t_1\right) \le 2\exp\left(-\frac{(deg(i))t_1^2}{2B^2}\right)$$

If $\left\|\sum_{j \in \mathcal{N}(i)} (\mathbf{x}_j - \mathbb{E}[\mathbf{x}_j])\right\|_2 \ge \sqrt{l}t_1$, then at least for one $k \in \{1, \ldots, l\}$, the inequality $\left|\sum_{j \in \mathcal{N}(i)} (\mathbf{x}_j[k] - \mathbb{E}[\mathbf{x}_j[k]])\right| \ge t_1$ holds. Hence, we have

$$\mathbb{P}\left(\left\|\sum_{j \in \mathcal{N}(i)} (\mathbf{x}_j - \mathbb{E}[\mathbf{x}_j])\right\|_2 \ge \sqrt{l}t_1\right) \le \mathbb{P}\left(\bigcup_{k=1}^{l}\left\{\left|\sum_{j \in \mathcal{N}(i)} (\mathbf{x}_j[k] - \mathbb{E}[\mathbf{x}_j[k]])\right| \ge t_1\right\}\right)$$

$$\le \sum_{k=1}^{l}\mathbb{P}\left(\left|\sum_{j \in \mathcal{N}(i)} (\mathbf{x}_j[k] - \mathbb{E}[\mathbf{x}_j[k]])\right| \ge t_1\right)$$

$$= 2 \cdot l \cdot \exp\left(-\frac{(deg(i))t_1^2}{2B^2}\right)$$

Let $t_1 = \frac{t_2}{\sqrt{l}}$, then we have

$$\mathbb{P}\left(\left\|\sum_{j \in \mathcal{N}(i)} (\mathbf{x}_j - \mathbb{E}[\mathbf{x}_j])\right\|_2 \geq t_2\right) \leq 2 \cdot l \cdot \exp\left(-\frac{(deg(i))t_2^2}{2B^2 l}\right)$$

Furthermore, we have

$$\|\mathbf{h}_i - \mathbb{E}[\mathbf{h}_i]\|_2 = \left\|\mathbf{W}\left(\sum_{j \in \mathcal{N}(i)} (\mathbf{x}_j - \mathbb{E}[\mathbf{x}_j])\right)\right\|_2$$

$$\leq \|\mathbf{W}\|_2 \left\|\sum_{j \in \mathcal{N}(i)} (\mathbf{x}_j - \mathbb{E}[\mathbf{x}_j])\right\|_2$$

$$= \rho(\mathbf{W}) \left\|\sum_{j \in \mathcal{N}(i)} (\mathbf{x}_j - \mathbb{E}[\mathbf{x}_j])\right\|_2,$$

where $\|\mathbf{W}\|_2$ is the matrix 2-norm of $\mathbf{W}$. Note that the last line uses the identity $\|\mathbf{W}\|_2 = \rho(\mathbf{W})$.

Then, for any $t > 0$, we have

$$\mathbb{P}(\|\mathbf{h}_i - \mathbb{E}[\mathbf{h}_i]\|_2 \geq t) \leq \mathbb{P}\left(\rho(\mathbf{W}) \left\|\sum_{j \in \mathcal{N}(i)} (\mathbf{x}_j - \mathbb{E}[\mathbf{x}_j])\right\|_2 \geq t\right)$$

$$= \mathbb{P}\left(\left\|\sum_{j \in \mathcal{N}(i)} (\mathbf{x}_j - \mathbb{E}[\mathbf{x}_j])\right\|_2 \geq \frac{t}{\rho(\mathbf{W})}\right)$$

$$\leq 2 \cdot l \cdot \exp\left(-\frac{(deg(i))t^2}{2\rho^2(\mathbf{W})B^2 l}\right),$$

which completes the proof. $\qquad\qquad\qquad\qquad\qquad\qquad\qquad\qquad\qquad\qquad\qquad\qquad\qquad$ $\square$

## B  PROOF OF THEOREM 2

**Theorem 2.** *Consider a graph $\mathcal{G} \sim CSBM(\boldsymbol{\mu}_1, \boldsymbol{\mu}_2, p, q)$. For any node $i$ in this graph, the linear classifier defined by the decision boundary $\mathcal{P}$ has a lower probability to mis-classify $\mathbf{h}_i$ than $\mathbf{x}_i$ when $deg(i) > (p+q)^2/(p-q)^2$.*

*Proof.* We only prove for nodes from classes $c_0$ since the case for nodes from classes $c_1$ is symmetric and the proof is exactly the same. For a node $i \in \mathcal{C}_0$, we have the follows

$$\mathbb{P}(\mathbf{x}_i \text{ is mis-classified}) = \mathbb{P}(\mathbf{w}^\top \mathbf{x}_i + \mathbf{b} \leq 0) \text{ for } i \in \mathcal{C}_0 \tag{7}$$

$$\mathbb{P}(\mathbf{h}_i \text{ is mis-classified}) = \mathbb{P}(\mathbf{w}^\top \mathbf{h}_i + \mathbf{b} \leq 0) \text{ for } i \in \mathcal{C}_0, \tag{8}$$

where $\mathbf{w}$ and $\mathbf{b} = -\mathbf{w}^\top (\boldsymbol{\mu}_1 + \boldsymbol{\mu}_1)/2$ is the parameters of the decision boundary $\mathcal{P}$. we have that

$$\mathbb{P}(\mathbf{w}^\top \mathbf{h}_i + \mathbf{b} \leq 0) = \mathbb{P}(\mathbf{w}^\top \sqrt{deg(i)}\mathbf{h}_i + \sqrt{deg(i)}\mathbf{b} \leq 0). \tag{9}$$

We denote the scaled version of $\mathbf{h}_i$ as $\mathbf{h}_i' = \sqrt{deg(i)}\mathbf{h}_i$. Then, $\mathbf{h}_i'$ follows

$$\mathbf{h}_i' = \sqrt{deg(i)}\mathbf{h}_i \sim N\left(\frac{\sqrt{deg(i)}(p\boldsymbol{\mu}_0 + q\boldsymbol{\mu}_1)}{p+q}, \mathbf{I}\right), \text{ for } i \in \mathcal{C}_0. \tag{10}$$

Because of the scale in Eq. (9), the decision boundary for $\mathbf{h}_i'$ is correspondingly moved to $\mathbf{w}^\top \mathbf{h}' + \sqrt{deg(i)}\mathbf{b} = 0$. Now, since $\mathbf{x}_i$ and $\mathbf{h}_i'$ share the same variance, to compare the mis-classification probabilities, we only need to compare the distance from their expected value to their corresponding decision boundary. Specifically, the two distances are as follows:

$$dis_{\mathbf{x}_i} = \frac{\|\boldsymbol{\mu}_0 - \boldsymbol{\mu}_1\|_2}{2}$$

$$dis_{\mathbf{h}_i'} = \frac{\sqrt{deg(i)}|p - q|}{(p+q)} \cdot \frac{\|\boldsymbol{\mu}_0 - \boldsymbol{\mu}_1\|_2}{2}. \tag{11}$$

The larger the distance is the smaller the mis-classification probability is. Hence, when $dis_{\mathbf{h}_i'} < dis_{\mathbf{x}_i}$, $\mathbf{h}_i'$ has a lower probability to be mis-classified than $\mathbf{x}_i$. Comparing the two distances, we conclude that when $deg(i) > \left(\frac{p+q}{p-q}\right)^2$, $\mathbf{h}_i'$ has a lower probability to be mis-classified than $\mathbf{x}_i$. Together with Eq. 9, we have that

$$\mathbb{P}(\mathbf{h}_i \text{ is mis-classified}) < \mathbb{P}(\mathbf{x}_i \text{ is mis-classified}) \text{ if } deg(i) > \left(\frac{p+q}{p-q}\right)^2, \tag{12}$$

which completes the proof.

$\square$

## C ADDITIONAL DETAILS AND RESULTS FOR SECTION 3.3.1

### C.1 DETAILS ON THE GENERATED GRAPHS

In this subsection, we present the details of the graphs that we generate in Section 3.3.1. Specifically, we detail the distributions $\{\mathcal{D}_c, c \in \mathcal{C}\}$ used in the examples, the number of added edges $K$, and the homophily ratio $h$. We provide the details for Cora and Citeseer in the following subsections. Note that the choices of distributions shown here are for illustrative purposes, to coincide with Observations 1 and 2. We adapted circulant matrix-like designs due to their simplicity.

#### C.1.1 CORA

There are 7 labels, which we denote as $\{0, 1, 2, 3, 4, 5, 6\}$. The distributions $\{\mathcal{D}_c, c \in \mathcal{C}\}$ are listed as follows. The values of $K$ and the homophily ratio of their corresponding generated graphs are shown in Table 3.

$$\mathcal{D}_0 : \mathsf{Categorical}([0, 0.5, 0, 0, 0, 0, 0.5]),$$
$$\mathcal{D}_1 : \mathsf{Categorical}([0.5, 0, 0.5, 0, 0, 0, 0]),$$
$$\mathcal{D}_2 : \mathsf{Categorical}([0, 0.5, 0, 0.5, 0, 0, 0]),$$
$$\mathcal{D}_3 : \mathsf{Categorical}([0, 0, 0.5, 0, 0.5, 0, 0]),$$
$$\mathcal{D}_4 : \mathsf{Categorical}([0, 0, 0, 0.5, 0, 0.5, 0]),$$
$$\mathcal{D}_5 : \mathsf{Categorical}([0, 0, 0, 0, 0.5, 0, 0.5]),$$
$$\mathcal{D}_6 : \mathsf{Categorical}([0.5, 0, 0, 0, 0, 0.5, 0]).$$

Table 3: # of added edges ($K$) and homophily ratio ($h$) values for generated graphs based on Cora.

| $K$ | 1003 | 2006 | 3009 | 4012 | 6018 | 8024 | 10030 | 12036 | 16048 | 20060 | 24072 |
|---|---|---|---|---|---|---|---|---|---|---|---|
| $h$ | 0.740 | 0.681 | 0.630 | 0.587 | 0.516 | 0.460 | 0.415 | 0.378 | 0.321 | 0.279 | 0.247 |

#### C.1.2 CITESEER

There are 6 labels, which we denote as $\{0, 1, 2, 3, 4, 5\}$. The distributions $\{\mathcal{D}_c, c \in \mathcal{C}\}$ are listed as follows. The values of $K$ and the homophily ratio of their corresponding generated graphs are shown in Table 4.

$$\mathcal{D}_0 : \mathsf{Categorical}([0, 0.5, 0, 0, 0, 0.5]),$$
$$\mathcal{D}_1 : \mathsf{Categorical}([0.5, 0, 0.5, 0, 0, 0]),$$
$$\mathcal{D}_2 : \mathsf{Categorical}([0, 0.5, 0, 0.5, 0, 0]),$$
$$\mathcal{D}_3 : \mathsf{Categorical}([0, 0, 0.5, 0, 0.5, 0]),$$
$$\mathcal{D}_4 : \mathsf{Categorical}([0, 0, 0, 0.5, 0, 0.5]),$$
$$\mathcal{D}_5 : \mathsf{Categorical}([0.5, 0, 0, 0, 0.5, 0]).$$

Table 4: # of added edges ($K$) and homophily ratio ($h$) values for generated graphs based on `Citeseer`

| $K$ | 1204 | 2408 | 3612 | 4816 | 7224 | 9632 | 12040 | 14448 | 19264 | 24080 | 28896 |
|---|---|---|---|---|---|---|---|---|---|---|---|
| $h$ | 0.650 | 0.581 | 0.527 | 0.481 | 0.410 | 0.357 | 0.317 | 0.284 | 0.236 | 0.202 | 0.176 |

### C.2 RESULTS ON MORE DATASETS: CHAMELEON AND SQUIRREL

We conduct similar experiments as those in Section 3.3.1 based on `Chameleon` and `Squirrel`. Note that both `Squirrel` and `Chameleon` have 5 labels, which we denote as $\{0, 1, 2, 3, 4\}$. We pre-define the same distributions for them as listed as follows. The values of $K$ and the homophily ratio of their corresponding generated graphs based on `Squirrel` and `Chameleon` are shown in Table 5 and Table 6, respectively.

$$\mathcal{D}_0 : \mathsf{Categorical}([0, 0.5, 0, 0, 0.5]),$$
$$\mathcal{D}_1 : \mathsf{Categorical}([0.5, 0, 0.5, 0, 0]),$$
$$\mathcal{D}_2 : \mathsf{Categorical}([0, 0.5, 0, 0.5, 0]),$$
$$\mathcal{D}_3 : \mathsf{Categorical}([0, 0, 0.5, 0, 0.5]),$$
$$\mathcal{D}_4 : \mathsf{Categorical}([0.5, 0, 0, 0.5, 0]).$$

Table 5: # of added edges ($K$) and homophily ratio ($h$) values for generated graphs based on `Squirrel`.

| $K$ | 12343 | 24686 | 37030 | 49374 | 61716 | 74060 | 86404 | 98746 | 111090 | 12434 | 135776 |
|---|---|---|---|---|---|---|---|---|---|---|---|
| $h$ | 0.215 | 0.209 | 0.203 | 0.197 | 0.192 | 0.187 | 0.182 | 0.178 | 0.173 | 0.169 | 0.165 |

Table 6: # of added edges ($K$) and homophily ratio ($h$) values for generated graphs based on `Chameleon`.

| $K$ | 1932 | 3866 | 5798 | 7730 | 9964 | 11596 | 13528 | 15462 | 17394 | 19326 | 21260 |
|---|---|---|---|---|---|---|---|---|---|---|---|
| $h$ | 0.223 | 0.217 | 0.210 | 0.205 | 0.199 | 0.194 | 0.189 | 0.184 | 0.180 | 0.176 | 0.172 |

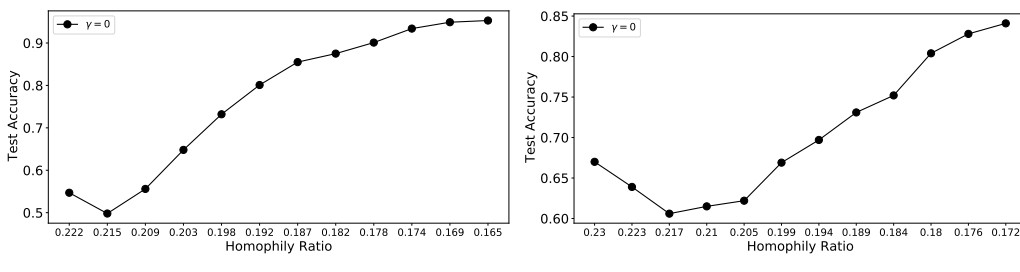

(a) Synthetic graphs generated from `Squirrel`  (b) Synthetic graphs generated from `Chameleon`

Figure 6: Performance of GCN on synthetic graphs with various homophily ratio.

The performance of the GCN model on these two sets of graphs (generated from `Squirrel` and `Chameleon`) is shown in Figure 6. The observations are similar to what we found for those generated graphs based on `Cora` and `Citeseer` in Section 3.3.1. Note that the original `Squirrel` and `Chameleon` graphs already have very low homophily, but we still observe a $V$-shape from the figures. This is because there are some neighborhood patterns in the original graphs, which are distinct from those that we designed for addition. Hence, when we add edges in the early stage, the performance decreases. As we add more edges, the designed pattern starts to mask the original patterns and the performance starts to increase.

## C.3 GCN's Performance in the Limit (as $K \to \infty$)

In this subsection, we illustrate that as $K \to \infty$, the accuracy of the GCN model approaches $100\%$. Specifically, we set $K$ to a set of larger numbers as listed in Table 7. Ideally, when $K \to \infty$, the homophily ratio will approach $0$ and the model performance will approach $100\%$ (for diverse-enough $\mathcal{D}_c$). The performance of the GCN model on the graphs described in Table 3 and Table 7 are shown in Figure 7. Clearly, the performance of the GCN model approaches the maximum as we continue to increase $K$.

Table 7: Extended # of added edges ($K$) and homophily ratio ($h$) values for generated graphs based on Cora.

| $K$ | 28084 | 32096 | 36108 | 40120 | 44132 | 48144 | 52156 | 56168 | 80240 |
|---|---|---|---|---|---|---|---|---|---|
| $h$ | 0.272 | 0.248 | 0.228 | 0.211 | 0.196 | 0.183 | 0.172 | 0.162 | 0.120 |

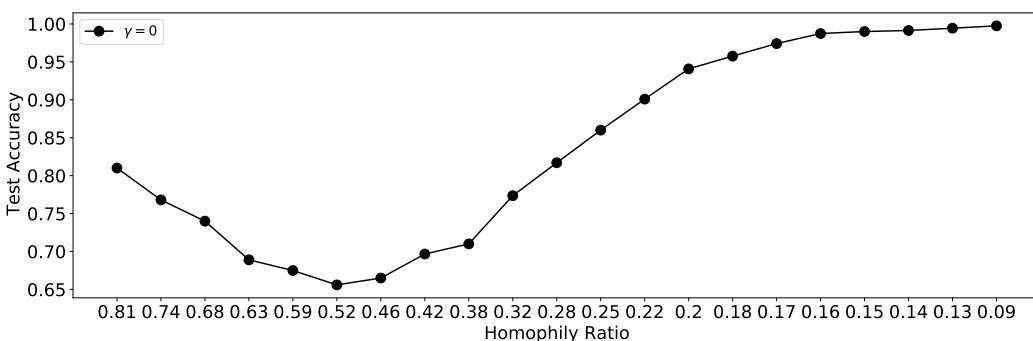

Figure 7: As the number of edges approaches $K \to \infty$, the homophily ratio $h \to 0$, and GCN's performance approaches $100\%$.

## C.4 Details of Algorithm 2

The pseudo code to describe the process to generate graphs with a noise level $\gamma$ is shown in Algorithm 2. The only difference from Algorithm 1 is in Line 6-7, we randomly add edges if the generated random number $r$ is smaller than the pre-defined $\gamma$ (with probability $\gamma$).

---

**Alg. 2:** Heterophilous Edge Addition with Noise

**input** : $\mathcal{G} = \{\mathcal{V}, \mathcal{E}\}, K, \{\mathcal{D}_c\}_{c=0}^{|\mathcal{C}|-1}$ and $\{\mathcal{V}_c\}_{c=0}^{|\mathcal{C}|-1}$
**output:** $\mathcal{G}' = \{\mathcal{V}, \mathcal{E}'\}$
Initialize $\mathcal{G}' = \{\mathcal{V}, \mathcal{E}\}, k = 1$ ;
**while** $1 \leq k \leq K$ **do**
    Sample node $i \sim \mathsf{Uniform}(\mathcal{V})$;
    Obtain the label, $y_i$ of node $i$;
    Sample a number $r \sim \mathsf{Uniform}(0,1)$ ;      // `Uniform(0,1)` denotes the continuous standard uniform distribution
    **if** $r \leq \gamma$ **then**
        Sample a label $c \sim \mathsf{Uniform}(\mathcal{C} \setminus \{y_i\})$;
    **else**
        Sample a label $c \sim \mathcal{D}_{y_i}$;
    Sample node $j \sim \mathsf{Uniform}(\mathcal{V}_c)$;
    Update edge set $\mathcal{E}' = \mathcal{E}' \cup \{(i,j)\}$;
    $k \leftarrow k + 1$;
**return** $\mathcal{G}' = \{\mathcal{V}, \mathcal{E}'\}$

---

# D    EXPERIMENTAL DETAILS: DATASETS, MODELS, AND RESULTS

We compare the standard GCN model (Kipf and Welling, 2016) with several recently proposed methods specifically designed for heterophilous graphs including H2GCN (Zhu et al., 2020b), GPR-GNN (Chien et al., 2021), and CPGNN (Zhu et al., 2020a). A brief introduction of these methods can be found in Appendix D.2.

## D.1    DATASETS

We give the number of nodes, edges, homophily ratios and distinct classes of datasets we used in this paper in Table 8.

Table 8: Benchmark dataset summary statistics.

|  | Cora | Citeseer | Pubmed | Chameleon | Squirrel | Actor | Cornell | Wisconsin | Texas |
|---|---|---|---|---|---|---|---|---|---|
| # Nodes ($|\mathcal{V}|$) | 2708 | 3327 | 19717 | 2277 | 5201 | 7600 | 183 | 251 | 183 |
| # Edges ($|\mathcal{E}|$) | 5278 | 4676 | 44327 | 31421 | 198493 | 26752 | 280 | 466 | 295 |
| Homophily Ratio ($h$) | 0.81 | 0.74 | 0.80 | 0.23 | 0.22 | 0.22 | 0.3 | 0.21 | 0.11 |
| # Classes ($|\mathcal{C}|$) | 7 | 6 | 3 | 5 | 5 | 5 | 5 | 5 | 5 |

## D.2    MODELS

- H2GCN (Zhu et al., 2020b) specifically designed several architectures to deal with heterophilous graphs, which include ego- and neighbor-embedding separation (skip connection), aggregation from higher-order neighborhoods, and combination of intermediate representations. We include two variants H2GCN-1 and H2GCN-2 with 1 or 2 steps of aggregations, respectively. We adopt the code published by the authors at `https://github.com/GemsLab/H2GCN`.
- GPR-GNN (Chien et al., 2021) performs feature aggregation for multiple steps and then linearly combines the features aggregated with different steps. The weights of the linear combination are learned during the model training. Note that it also includes the original features before aggregation in the combination. We adopt the code published by the authors at `https://github.com/jianhao2016/GPRGNN`.
- CPGNN (Zhu et al., 2020a) incorporates the label compatibility matrix to capture the connection information between classes. We adopted two variants of CPGNN that utilize MLP and ChebyNet (Defferrard et al., 2016) as base models to pre-calculate the compatibility matrix, respectively. We use two aggregation layers for both variants. We adopt the code published by the authors at `https://github.com/GemsLab/CPGNN`.

## D.3    MLP+GCN

We implement a simple method to linearly combine the learned features from the GCN model and an MLP model. Let $\mathbf{H}_{GCN}^{(2)} \in \mathbb{R}^{|\mathcal{V}| \times |\mathcal{C}|}$ denote the output features from a 2-layer GCN model, where $|\mathcal{V}|$ and $|\mathcal{C}|$ denote the number of nodes and the number of classes, respectively. Similarly, we use $\mathbf{H}_{MLP}^{(2)} \in \mathbb{R}^{|\mathcal{V}| \times |\mathcal{C}|}$ to denote the features output from a 2-layer MLP model. We then combine them for classification. The process can be described as follows.

$$\mathbf{H} = \alpha \cdot \mathbf{H}_{GCN}^{(2)} + (1 - \alpha) \cdot \mathbf{H}_{MLP}^{(2)}, \tag{13}$$

where $\alpha$ is a hyperparameter balancing the two components. We then apply a row-wise softmax to each row of $\mathbf{H}$ to perform the classification.

## D.4    PARAMETER TUNING AND RESOURCES USED

We tune parameters for GCN, GPR-GCN, CPGNN, and MLP+GCN from the following options:

- learning rate: $\{0.002, 0.005, 0.01, 0.05\}$
- weight decay $\{5e-04, 5e-05, 5e-06, 5e-07, 5e-08, 1e-05, 0\}$
- dropout rate: $\{0, 0.2, 0.5, 0.8\}$.

For GPR-GNN, we use the "PPR" as the initialization for the coefficients. For MLP+GCN, we tune $\alpha$ from $\{0.2, 0.4, 0.6, 0.8, 1\}$. Note that the parameter search range encompasses the range adopted in the original papers to avoid unfairness issues.

All experiments are run on a cluster equipped with *Intel(R) Xeon(R) CPU E5-2680 v4 @ 2.40GHz* CPUs and *NVIDIA Tesla K80* GPUs.

## D.5 MORE RESULTS

|  | Cora | Citeseer | Pubmed | Chameleon | Squirrel | Actor | Cornell | Wisconsin | Texas |
|---|---|---|---|---|---|---|---|---|---|
| GCN | $87.12 \pm 1.38$ | $76.50 \pm 1.61$ | $88.52 \pm 0.41$ | $67.96 \pm 1.82$ | $54.47 \pm 1.17$ | $30.31 \pm 0.98$ | $59.35 \pm 4.19$ | $61.76 \pm 6.15$ | $63.81 \pm 5.27$ |
| MLP | $75.04 \pm 1.97$ | $72.40 \pm 1.97$ | $87.84 \pm 0.30$ | $48.11 \pm 2.23$ | $31.68 \pm 1.90$ | $36.17 \pm 1.09$ | $\mathbf{84.86 \pm 6.04}$ | $86.29 \pm 4.50$ | $83.30 \pm 4.54$ |
| MLP + GCN | $87.01 \pm 1.35$ | $76.35 \pm 1.85$ | $\mathbf{89.77 \pm 0.39}$ | $\mathbf{68.04 \pm 1.86}$ | $\mathbf{54.48 \pm 1.11}$ | $\mathbf{36.24 \pm 1.09}$ | $84.82 \pm 4.87$ | $86.43 \pm 4.00$ | $83.60 \pm 6.04$ |
| H2GCN-1 | $86.92 \pm 1.37$ | $77.07 \pm 1.64$ | $89.40 \pm 0.34$ | $57.11 \pm 1.58$ | $36.42 \pm 1.89$ | $35.86 \pm 1.03$ | $82.16 \pm 6.00$ | $\mathbf{86.67 \pm 4.69}$ | $\mathbf{84.86 \pm 6.77}$ |
| H2GCN-2 | $\mathbf{87.81 \pm 1.35}$ | $\mathbf{76.88 \pm 1.77}$ | $89.59 \pm 0.33$ | $59.39 \pm 1.98$ | $37.90 \pm 2.02$ | $35.62 \pm 1.30$ | $82.16 \pm 6.00$ | $85.88 \pm 4.22$ | $82.16 \pm 5.28$ |
| CPGNN-MLP | $85.84 \pm 1.20$ | $74.80 \pm 0.92$ | $86.58 \pm 0.37$ | $54.53 \pm 2.37$ | $29.13 \pm 1.57$ | $35.76 \pm 0.92$ | $79.93 \pm 6.12$ | $84.58 \pm 2.72$ | $82.62 \pm 6.88$ |
| CPGNN-Cheby | $87.23 \pm 1.31$ | $76.64 \pm 1.43$ | $88.41 \pm 0.33$ | $65.17 \pm 3.17$ | $29.25 \pm 4.17$ | $34.28 \pm 0.77$ | $75.08 \pm 7.51$ | $79.19 \pm 2.80$ | $75.96 \pm 5.66$ |
| GPR-GNN | $86.79 \pm 1.27$ | $75.55 \pm 1.56$ | $86.79 \pm 0.55$ | $66.31 \pm 2.05$ | $50.56 \pm 1.51$ | $33.94 \pm 0.95$ | $79.27 \pm 6.03$ | $83.73 \pm 4.02$ | $84.43 \pm 4.10$ |

# E  HEATMAPS FOR OTHER BENCHMARKS

We provide the heatmaps for `Citeseer` and `Pubmed` in Figure 8 and those for `Squirrel`, `Texas`, and `Wisconsin` in Figure 9. For the `Citeseer` and `Pubmed`, which have high homophily, the observations are similar to those of `Cora` as we described in Section 4.2. For `Squirrel`, there are some patterns; the intra-class similarity is generally higher than inter-class similarities. However, these patterns are not very strong, i.e, the differences between them are not very large, which means that the neighborhood patterns of different labels are not very distinguishable from each other. This substantiates the middling performance of GCN on `Squirrel`. Both `Texas` and `Wisconsin` are very small, with 183 nodes, 295 edges and 251 nodes, 466 edges, respectively. The average degree is extremely small ($< 2$). Hence, the similarities presented in the heatmap may present strong bias. Especially, in `Texas`, there is only 1 node with label 1. In `Wisconsin`, there are only 10 nodes with label 0.

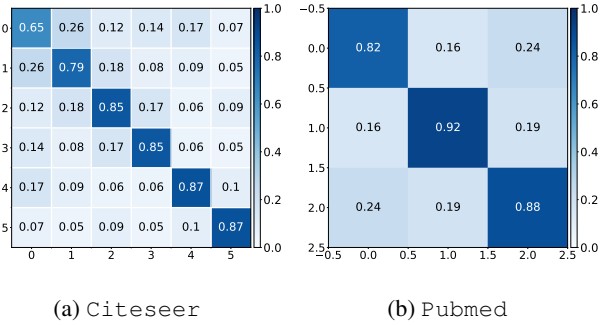

(a) `Citeseer`  (b) `Pubmed`

Figure 8: Cross-class neighborhood similarity on `Citeseer` and `Pubmed`. On both graphs, the intra-class similarity is clearly higher than the inter-class ones.

# F  EXTENDING THEOREM 2 TO MULTIPLE CLASSES

Below, we provide a proof sketch and illustration for an extension to Theorem 2's main results in a multi-class context. Specifically, we consider a special case for more tractable analysis.

Consider a $K$-class CSBM. The nodes in the generated graphs consist of $K$ disjoint sets of the same size $\mathcal{C}_1, \ldots, \mathcal{C}_K$ corresponding to the $K$ classes, respectively. Edges are generated according to an intra-class probability $p$ and an inter-class probability $q$. Specifically, for any two nodes in the graph, if they are from the same class, then an edge is generated to connect them with probability $p$, otherwise, the probability is $q$. For each node $i$, its initial associated features $\mathbf{x}_i \in \mathbb{R}^l$ are sampled from a Gaussian distribution $\mathbf{x}_i \sim N(\boldsymbol{\mu}, \mathbf{I})$, where $\boldsymbol{\mu} = \boldsymbol{\mu}_k \in \mathbb{R}^l$ for $i \in \mathcal{C}_k$ with $k \in \{1, \ldots, K\}$ and $\boldsymbol{\mu}_z \neq \boldsymbol{\mu}_w \forall z, w \in \{1, \ldots, K\}$. We further assume that the distance between the mean of distributions corresponding to any two classes is equivalent, i.e, $\|\mathbf{u}_z - \mathbf{u}_w\|_2 = D, \ \forall z, w \in \{1, \ldots, K\}$, where $D$ is a positive constant. We illustrate the 3-classes case in Figure 10, where we use the dashed circles to demonstrate the standard deviation. Note that the $K$ classes of the CSBM model are symmetric to each other. Hence, the optimal decision boundary of the $K$ classes are defined by a set

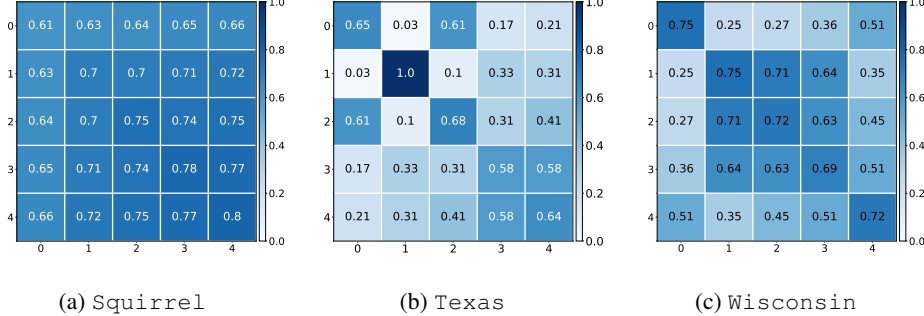

(a) Squirrel          (b) Texas          (c) Wisconsin

Figure 9: Cross-class neighborhood similarity on `Squirrel`, `Texas` and `Wisconsin`. The inter-class similarity on `Squirrel` is slightly higher than intra-class similarity for most classes, which substantiates the middling performance of GCN. Both `Texas` and `Wisconsin` are quite small, hence the cross-class similarity in these two graphs present severe bias and may not provide precise information about these graphs.

of $\binom{K}{2}$ hyperplanes (see Figure 10 for an example), where each hyperplane equivalently separates two classes. Similar to the analysis in the binary case (see the description below Proposition 1), for any given two classes $c_z$ and $c_w$, the hyperplane is $\mathcal{P} = \{\mathbf{x}|\mathbf{w}^\top\mathbf{x} - \mathbf{w}^\top(\boldsymbol{\mu}_z + \boldsymbol{\mu}_w)/2\}$ with $\mathbf{w} = (\mathbf{u}_z - \mathbf{u}_w)/\|\mathbf{u}_z - \mathbf{u}_w\|_2$, which is orthogonal to $(\boldsymbol{\mu}_z - \boldsymbol{\mu}_w)$ and going through $(\boldsymbol{\mu}_z + \boldsymbol{\mu}_w)/2$. For example, in Figure 10, the decision boundaries separate the entire space to 3 areas corresponding to the 3 classes. Each decision boundary is defined by a hyperplane equally separating two classes. For example, the descision bounadry between $c_1$ and $c_2$ is orthogonal to $(\boldsymbol{\mu}_1 - \boldsymbol{\mu}_2)$ and going through the middle point $(\boldsymbol{\mu}_1 + \boldsymbol{\mu}_2)/2$. Clearly, the linear separability is dependent on the distance $D$ between the classes and also the standard deviations of each class' node features. More specifically, linear separability is favored by a larger distance $D$ and smaller standard deviation.

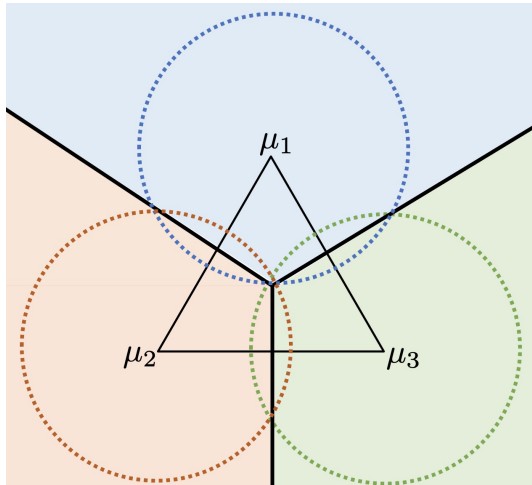

Figure 10: 3-class CSBM in 2-dimensional space. The dashed circles demonstrate the standard deviation for each class. The entire space is split to three areas (indicated by different colors) corresponding to the three classes for optimal decision making. The equilateral triangle indicates that the distance between the means of any two classes is the same.

Next, we discuss how the GCN operation affects the linear separability. Overall, we want to demonstrate the following: (i) after the GCN operation, the classes are still symmetric to each other; (ii) the GCN operation will reduce the distance between the classes (in terms of the expectation of the output embeddings) and reduce the standard deviation of the classes; and (iii) reducing the distance impairs the separability while reducing the standard deviation improves the separability. Hence, we

analyze the two effect and provide a threshold. We describe these three items in a more detailed way as follows.

- **Preservation of Symmetry.** For any class $c_z$, $z \in \{1, \ldots, K\}$, its neighborhood label distribution $\mathcal{D}_{c_z}$ can be described by a vector where only the $z$-th element equals $\frac{p}{p+(K-1)q}$ and all other elements equal to $\frac{q}{p+(K-1)q}$. We consider the aggression process $\mathbf{h}_i = \frac{1}{deg(i)} \sum_{j \in \mathcal{N}(i)} \mathbf{x}_j$. Then, for a node $i$ with label $c_z$, its features obtained after this process follow the following Gaussian distribution.

$$\mathbf{h}_i \sim N \left( \frac{p\boldsymbol{\mu}_z + \sum\limits_{k \in \{1,\ldots K\}, k \neq z} q\boldsymbol{\mu}_k}{p + (K-1)q}, \frac{\mathbf{I}}{\sqrt{deg(i)}} \right), \text{for } i \in \mathcal{C}_z \tag{14}$$

Specifically, we denote the expectation of class $c_z$ after the GCN operation as $\mathbb{E}_{c_z}[\mathbf{h}] = \frac{p\boldsymbol{\mu}_z + \sum\limits_{k \in \{1,\ldots K\}, k \neq z} q\boldsymbol{\mu}_k}{p+(K-1)q}$. We next show that the distance between any two classes is the same. Specifically, for any two classes $c_z$ and $c_w$, after the GCN operation, the distance between their expectation is as follows.

$$\|\mathbb{E}_{c_z}[\mathbf{h}] - \mathbb{E}_{c_k}[\mathbf{h}]\|_2 = \left\| \frac{p\boldsymbol{\mu}_z + \sum\limits_{k \in \{1,\ldots K\}, k \neq z} q\boldsymbol{\mu}_k}{p + (K-1)q} - \frac{p\boldsymbol{\mu}_w + \sum\limits_{k \in \{1,\ldots K\}, k \neq w} q\boldsymbol{\mu}_k}{p + (K-1)q} \right\|_2$$

$$= \left\| \frac{(p-q)(\boldsymbol{\mu}_z - \boldsymbol{\mu}_w)}{p + (K-1)q} \right\|_2 = \frac{|p-q|}{p+(K-1)q} \|\boldsymbol{\mu}_z - \boldsymbol{\mu}_w\|_2 \tag{15}$$

Note that we have $\|\boldsymbol{\mu}_z - \boldsymbol{\mu}_w\|_2 = D$ for any pair of classes $c_z$ and $c_w$. Hence, after the GCN operation, the distance between any two classes is still the same. Thus, the classes are symmetric two each other.

- **Reducing inter-class distance and intra-class standard deviation.** According to Eq (15), after the graph convolution operation, the distance between any two classes is reduced by a factor of $\frac{|p-q|}{p+(K-1)q}$. On the other hand, according to Eq. (14), the standard deviation depends on the degree of nodes. Specifically, for node $i$ with degree $deg(i)$, its standard deviation is reduced by a factor of $\sqrt{deg(i)}$.

- **Implications for separability.** For a node $i$, the probability of being mis-classified is the probability of its embedding falling out of its corresponding decision area. This probability depends on both the inter-class distance between classes (in terms of means) and the intra-class standard deviation. To compare the linear separability before and after the graph convolution operation, we scale the distance between classes before and after the graph convolution operation to be the same. Specifically, we scale $\mathbf{h}_i$ as $\mathbf{h}'_i = \frac{p+(K-1)q}{|p-q|} \mathbf{h}_i$. Note that classifying $\mathbf{h}_i$ is equivalent to classifying $\mathbf{h}'_i$. Based on $\mathbf{h}'$, the distance between any two classes $c_z$, $c_w$ is scaled to $\|\boldsymbol{\mu}_z - \boldsymbol{\mu}_w\|_2 = D$, which is equivalent to the class distance before the graph convolution operation. Correspondingly, the standard deviation for $\mathbf{h}'_i$ equals to $\frac{p+(K-1)q}{|p-q|} \cdot \frac{\mathbf{I}}{\sqrt{deg(i)}}$. Now, to compare the mis-classification probability before and after the graph convolution, we only need to compare the standard deviations as the distances between classes in these two scenarios has been scaled to the same. More specifically, when $\frac{p+(K-1)q}{|p-q|} \cdot \frac{\mathbf{I}}{\sqrt{deg(i)}} < 1$, the mis-classification probability for node $i$ is reduced after the GCN model, otherwise, the mis-classification probability is increased after the GCN model. In other words, for nodes with degree larger than $\frac{(p+(K-1)q)^2}{(p-q)^2}$, the mis-classification rate can be reduced after the graph convolution operation. This thereold is similar to the one we developed in Theorem 2 and similar analysis/discussions as those for Theorem 2 follows for the this multiple-class case.

## G    OTHER METRICS THAN COSINE SIMILARITY

The choice of similarity measure would not affect the results and conclusions significantly. We empirically demonstrate this argument by investigating two other metrics: Euclidean distance and Hellinger distance. The heatmaps based on these two metrics for Chamelon dataset is shown in Figure 11 in Appendix G. The patterns demonstrated in these two heatmaps are similar to those

observed in Figure 5(b), where cosine similarity is adopted. Note that larger distance means lower similarity. Hence, the numbers in Figure 11 should be interpreted in the opposite way as those in Figure 5.

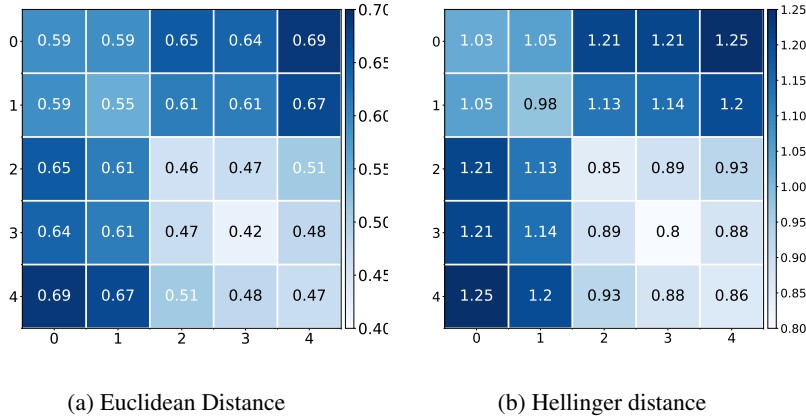

(a) Euclidean Distance        (b) Hellinger distance

Figure 11: Cross-class neighborhood patterns on `Chameleon` based on different metrics than cosine similarity metric.

# H   ADDITIONAL EXPERIMENTS WITH DIFFERENT SETTINGS

In this section, we discuss additional experiments for Section 3.3.1.

## H.1   GENERATING GRAPHS WITH OTHER NEIGHBORHOOD DISTRIBUTIONS

In this section, we extend the experiments in Section 3.3.1 by including more patterns for neighborhood distributions. We aim to illustrate if the neighborhood distribution for different classes (labels) are sufficiently distinguishable from each other, we should generally observe similar $V$-shape curves as in Figure 3 in Section 3.3.1. However, it is impractical to enumerate all possible neighborhood distributions. Hence, in this section, we include two additional neighborhood distribution patterns in Section H.1.1 and two extreme patterns in Section H.1.2.

### H.1.1   ADDITIONAL PATTERNS

Here, for both `Cora` and `Citeseer`, we adopt two additional sets of neighborhood distributions for adding new edges. For convenience, for `Cora`, we name the two neighborhood distribution patterns as *Cora Neighborhood Distribution Pattern 1* and *Cora Neighborhood Distribution Pattern 2*. Similarly, for `Citeseer`, we name the two neighborhood distribution patterns as *Citeseer Neighborhood Distribution Pattern 1* and *Citeseer Neighborhood Distribution Pattern 2*. We follow Algorithm 1 to generate graphs while utilizing these neighborhood distributions as the $\{\mathcal{D}_c\}_{c=0}^{|C|-1}$ for Algorithm 1.

The two neighborhood distribution patterns for `Cora` are listed as bellow. Figure 12 and Figure 13 show GCN's performance on graphs generated from `Cora` following *Cora Neighborhood Distribution Pattern 1* and *Cora Neighborhood Distribution Pattern 2*, respectively.

*Cora Neighborhood Distribution Pattern 1*:

$$\mathcal{D}_0 : \mathsf{Categorical}([0, \frac{1}{3}, \frac{1}{3}, \frac{1}{3}, 0, 0, 0]),$$

$$\mathcal{D}_1 : \mathsf{Categorical}([\frac{1}{3}, 0, 0, 0, \frac{1}{3}, \frac{1}{3}, 0]),$$

$$\mathcal{D}_2 : \mathsf{Categorical}([\frac{1}{3}, 0, 0, 0, 0, \frac{1}{3}, \frac{1}{3}]),$$

$$\mathcal{D}_3 : \mathsf{Categorical}([\frac{1}{3}, 0, 0, 0, \frac{1}{3}, 0, \frac{1}{3}]),$$

$$\mathcal{D}_4 : \mathsf{Categorical}([0, \frac{1}{3}, 0, \frac{1}{3}, 0, \frac{1}{3}, 0]),$$

$$\mathcal{D}_5 : \mathsf{Categorical}([0, \frac{1}{3}, \frac{1}{3}, 0, \frac{1}{3}, 0, 0]),$$

$$\mathcal{D}_6 : \mathsf{Categorical}([0, 0, \frac{1}{2}, \frac{1}{2}, 0, 0, 0]).$$

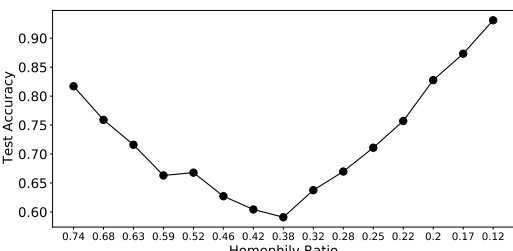

Figure 12: Performance of GCN on synthetic graphs from `Cora`. Graphs are generated following *Cora Neighborhood Distribution Pattern 1*

*Cora Neighborhood Distribution Pattern 2*:

$$\mathcal{D}_0 : \mathsf{Categorical}([0, \frac{1}{2}, 0, \frac{1}{2}, 0, 0, 0]),$$

$$\mathcal{D}_1 : \mathsf{Categorical}([\frac{1}{2}, 0, 0, \frac{1}{2}, 0, 0, 0]),$$

$$\mathcal{D}_2 : \mathsf{Categorical}([0, 0, 0, 1, 0, 0, 0]),$$

$$\mathcal{D}_3 : \mathsf{Categorical}([\frac{1}{5}, \frac{1}{5}, \frac{1}{5}, 0, \frac{1}{5}, \frac{1}{5}], 0),$$

$$\mathcal{D}_4 : \mathsf{Categorical}([0, 0, 0, \frac{1}{2}, 0, \frac{1}{2}, 0]),$$

$$\mathcal{D}_5 : \mathsf{Categorical}([0, \frac{1}{3}, \frac{1}{3}, 0, \frac{1}{3}, 0, 0]),$$

$$\mathcal{D}_6 : \mathsf{Categorical}([0, 0, 0, 0, 0, 0, 1]).$$

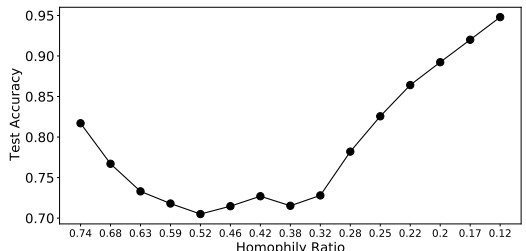

Figure 13: Performance of GCN on synthetic graphs from `Cora`. Graphs are generated following *Cora Neighborhood Distribution Pattern 2*

Clearly, the results demonstrated in Figure 12 and Figure 13 are similar to the black curve ($\gamma=0$) in 3a. More specifically, they present *V*-shape curves.

The two neighborhood distribution patterns for `Citeseer` are listed as bellow. Figure 14 and Figure 15 show GCN's performance on graphs generated from `Citeseer` following *Citeseer Neighborhood Distribution Pattern 1* and *Citeseer Neighborhood Distribution Pattern 2*, respectively.

*Citeseer Neighborhood Distribution Pattern 1*:

$$\mathcal{D}_0 : \mathsf{Categorical}([0, \frac{1}{2}, \frac{1}{2}, 0, 0, 0]),$$

$$\mathcal{D}_1 : \mathsf{Categorical}([\frac{1}{3}, 0, 0, 0, \frac{1}{3}, \frac{1}{3}]),$$

$$\mathcal{D}_2 : \mathsf{Categorical}([\frac{1}{2}, 0, 0, 0, 0, \frac{1}{2}]),$$

$$\mathcal{D}_3 : \mathsf{Categorical}([0, 0, 0, 0, 1, 0]),$$

$$\mathcal{D}_4 : \mathsf{Categorical}([0, \frac{1}{3}, 0, \frac{1}{3}, 0, \frac{1}{3}]),$$

$$\mathcal{D}_5 : \mathsf{Categorical}([0, \frac{1}{3}, \frac{1}{3}, 0, \frac{1}{3}, 0]),$$

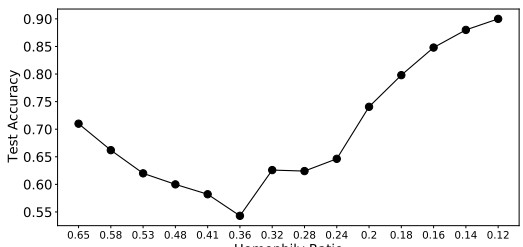

Figure 14: Performance of GCN on synthetic graphs from `Citeseer`. Graphs are generated following *Citeseer Neighborhood Distribution Pattern 1*

*Citeseer Neighborhood Distribution Pattern 2*:

$$\mathcal{D}_0 : \mathsf{Categorical}([0, \tfrac{1}{2}, 0, \tfrac{1}{2}, 0, 0]),$$

$$\mathcal{D}_1 : \mathsf{Categorical}([\tfrac{1}{2}, 0, 0, \tfrac{1}{32}, 0]),$$

$$\mathcal{D}_2 : \mathsf{Categorical}([0, 0, 0, 1, 0, 0]),$$

$$\mathcal{D}_3 : \mathsf{Categorical}([\tfrac{1}{5}, \tfrac{1}{5}, \tfrac{1}{5}, 0, \tfrac{1}{5}, \tfrac{1}{5}]),$$

$$\mathcal{D}_4 : \mathsf{Categorical}([0, 0, 0, \tfrac{1}{2}, 0, \tfrac{1}{2}]),$$

$$\mathcal{D}_5 : \mathsf{Categorical}([0, 0, 0, \tfrac{1}{2}, \tfrac{1}{2}, 0]),$$

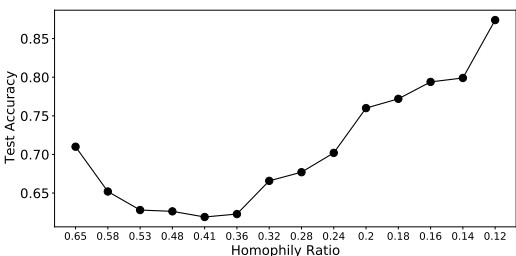

Figure 15: Performance of GCN on synthetic graphs from `Citeseer`. Graphs are generated following *Citeseer Neighborhood Distribution Pattern 2*

Clearly, the results demonstrated in Figure 14 and Figure 15 are similar to the black curve ($\gamma$=0) in 3b. More specifically, they present $V$-shape curves.

### H.1.2 EXTREME NEIGHBORHOOD DISTRIBUTION PATTERNS

In this subsection, we further extend the experiments in Section 3.3.1 by investigating "extreme" neighborhood distribution patterns suggested by WXGg. More specifically, these two patterns are "a given label are connected to a single different label" and "a given label are connected to all other labels other excluding its own label". For convenience, we denote these two types of neighborhood distributions as *single* and *all*, respectively. We utilize these neighborhood distributions as $\{\mathcal{D}_c\}_{c=0}^{|\mathcal{C}|-1}$ for generating synthetic graphs. **Next, we first present the results for `Cora` with analysis for both the *single* and *all* neighborhood distribution patterns. Then, we present the results for `Citeseer` but omit analysis and discussion since the observations are similar to those we make for `Cora`.**

Note that to ensure "a label is only connected to a single different label", we need to group the labels into pairs and "connect them". Since there are different ways to group labels into pairs, there exists various ways to formulate the *single* neighborhood distributions. We demonstrate one of the *single* neighborhood distributions as all the other possible are symmetric to each other. For `Cora`, the *single* neighborhood distribution we adopted is as follows. Note that there are 7 labels in `Cora`, and we could not pair all the labels. Thus, we leave one of labels (label 4 in our setting) untouched, i.e, it is not connect to other labels during the edge addition process.

$$\mathcal{D}_0 : \mathsf{Categorical}([0, 1, 0, 0, 0, 0, 0]),$$
$$\mathcal{D}_1 : \mathsf{Categorical}([1, 0, 0, 0, 0, 0, 0]),$$
$$\mathcal{D}_2 : \mathsf{Categorical}([0, 0, 0, 1, 0, 0, 0]),$$
$$\mathcal{D}_3 : \mathsf{Categorical}([0, 0, 1, 0, 0, 0, 0]),$$
$$\mathcal{D}_4 : \mathsf{Categorical}([0, 0, 0, 0, 0, 0, 0]),$$
$$\mathcal{D}_5 : \mathsf{Categorical}([0, 0, 0, 0, 0, 0, 1]),$$
$$\mathcal{D}_6 : \mathsf{Categorical}([0, 0, 0, 0, 0, 1, 0]).$$

The GCN's performance on these graphs generated from `Cora` following the *single* neighborhood distribution pattern is shown in Figure 16a. It clearly presents a $V$-shape curve. Almost perfect performance can be achieved when the homophily ratio gets close to 0. This is because the *single* neighborhood distributions for different labels are clearly distinguishable from each other. We further demonstrate the heatmap of cross-class similarity for the generated graph with homophily ratio 0.07 (the right most point in Figure 16a) in Figure 16b. Clearly, this graph has very high intra-class similarity and very low inter-class similarity, which explains the good performance.

For `Cora`, the *all* neighborhood distribution patterns can be described as follows.

$$\mathcal{D}_0 : \mathsf{Categorical}([0, \tfrac{1}{6}, \tfrac{1}{6}, \tfrac{1}{6}, \tfrac{1}{6}, \tfrac{1}{6}, \tfrac{1}{6}]),$$

$$\mathcal{D}_1 : \mathsf{Categorical}([\tfrac{1}{6}, 0, \tfrac{1}{6}, \tfrac{1}{6}, \tfrac{1}{6}, \tfrac{1}{6}, \tfrac{1}{6}]),$$

$$\mathcal{D}_2 : \mathsf{Categorical}([\tfrac{1}{6}, \tfrac{1}{6}, 0, \tfrac{1}{6}, \tfrac{1}{6}, \tfrac{1}{6}, \tfrac{1}{6}]),$$

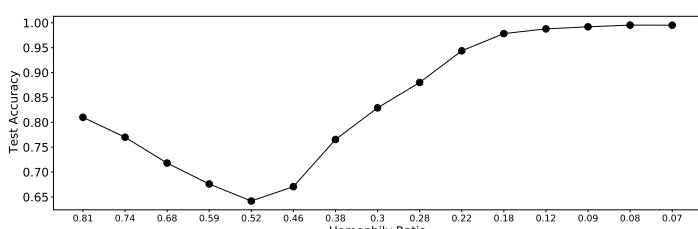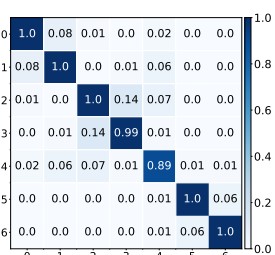

(a) Performance of GCN on synthetic graphs from `Cora`. Graphs are generated following *single* neighborhood distribution.

(b) Cross-class neighborhood similarity for the graph generated from `Cora` following *single* with homophily ratio 0.07 (the right most point in Figure 16a).

Figure 16: `Cora`: *single* neighborhood distribution pattern

$$\mathcal{D}_3 : \mathsf{Categorical}([\frac{1}{6}, \frac{1}{6}, \frac{1}{6}, 0, \frac{1}{6}, \frac{1}{6}, \frac{1}{6}]),$$

$$\mathcal{D}_4 : \mathsf{Categorical}([\frac{1}{6}, \frac{1}{6}, \frac{1}{6}, \frac{1}{6}, 0, \frac{1}{6}, \frac{1}{6}]),$$

$$\mathcal{D}_5 : \mathsf{Categorical}([\frac{1}{6}, \frac{1}{6}, \frac{1}{6}, \frac{1}{6}, \frac{1}{6}, 0, \frac{1}{6}]),$$

$$\mathcal{D}_6 : \mathsf{Categorical}([\frac{1}{6}, \frac{1}{6}, \frac{1}{6}, \frac{1}{6}, \frac{1}{6}, \frac{1}{6}, 0]).$$

The GCN's performance on these graphs generated from `Cora` following the *all* neighborhood distribution pattern is shown in Figure 17a. Again, it clearly presents a $V$-shape curve. However, the performance is not perfectly good even when we add extremely large number of edges. For example, for the right most point in Figure 17a, we add almost as 50 times many as edges into the graph and the homophily ratio is reduced to 0.03 while GCN's performance for it is only around 70%. This is because the *all* neighborhood distributions for different labels are not easily distinguishable from each other. More specifically, any two neighborhood distributions for different labels in *all* are very similar each other (they share "4 labels"). We further empirically demonstrate this by providing the heatmap of cross-class similarity for the generated graph with homophily ratio 0.03 (the right most point in Figure 17a) in Figure 17b. As per our discussion in Observation 2, this graph is with not such "good" heterophily and GCNs cannot produce perfect performance for it. This observation further demonstrates our key argument that the distinguishability of the distributions for different labels are important for performance.

For `Citeseer`, the *single* neighborhood distribution we adopted is as follows. The GCN's performance on these graphs generated from `Citeseer` following the *single* neighborhood distribution pattern is shown in Figure 18a. The heatmap of cross-class similarity for the generated graph with homophily ratio 0.06 (the right most point in Figure 18a) in Figure 18b.

$$\mathcal{D}_0 : \mathsf{Categorical}([0, 1, 0, 0, 0, 0]),$$
$$\mathcal{D}_1 : \mathsf{Categorical}([1, 0, 0, 0, 0, 0]),$$
$$\mathcal{D}_2 : \mathsf{Categorical}([0, 0, 0, 1, 0, 0]),$$
$$\mathcal{D}_3 : \mathsf{Categorical}([0, 0, 1, 0, 0, 0]),$$
$$\mathcal{D}_4 : \mathsf{Categorical}([0, 0, 0, 0, 0, 1]),$$
$$\mathcal{D}_5 : \mathsf{Categorical}([0, 0, 0, 0, 1, 0]).$$

For `Citeseer`, the *all* neighborhood distribution patterns can be described as follows. The GCN's performance on these graphs generated from `Citeseer` following the *all* neighborhood distribution pattern is shown in Figure 19a. The heatmap of cross-class similarity for the generated graph with homophily ratio 0.01 (the right most point in Figure 19a) in Figure 19b.

$$\mathcal{D}_0 : \mathsf{Categorical}([0, 1/5, 1/5, 1/5, 1/5, 1/5]),$$

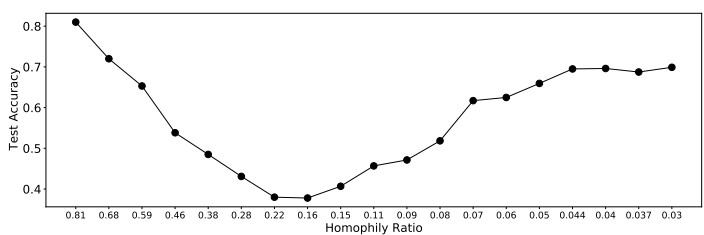 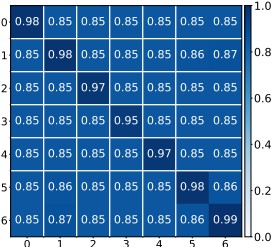

(a) Performance of GCN on synthetic graphs from `Cora`. Graphs are generated following *all* neighborhood distribution.

(b) Cross-class neighborhood similarity for the graph generated from `Cora` following *all* with homophily ratio 0.03 (the right most point in Figure 17a).

Figure 17: `Cora`: *all* neighborhood distribution pattern

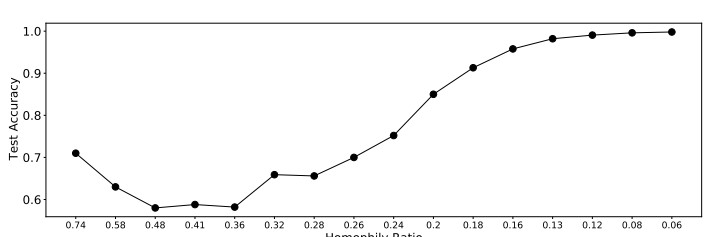 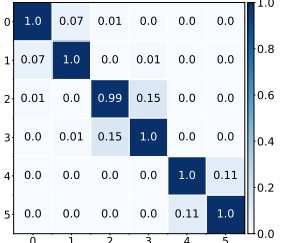

(a) Performance of GCN on synthetic graphs from `Citeseer`. Graphs are generated following *single* neighborhood distribution.

(b) Cross-class neighborhood similarity for the graph generated from `Citeseer` following *single* with homophily ratio 0.06.

Figure 18: `Citeseer`: *single* neighborhood distribution pattern

$$\mathcal{D}_1 : \text{Categorical}([1/5, 0, 1/5, 1/5, 1/5, 1/5]),$$
$$\mathcal{D}_2 : \text{Categorical}([1/5, 1/5, 0, 1/5, 1/5, 1/5]),$$
$$\mathcal{D}_3 : \text{Categorical}([1/5, 1/5, 1/5, 0, 1/5, 1/5]),$$
$$\mathcal{D}_4 : \text{Categorical}([1/5, 1/5, 1/5, 1/5, 0, 1/5]),$$
$$\mathcal{D}_5 : \text{Categorical}([1/5, 1/5, 1/5, 1/5, 1/5, 0]).$$

Similar observations as those we made for `Cora` can be made for `Citeseer`, hence we do not repeat the analysis here.

## I  LIMITATION

Though we provide new perspectives and understandings of GCN's performance on heterophilous graphs, our work has some limitations. To make the theoretical analysis more feasible, we make a few assumptions. We dropped the non-linearity in the analysis since the main focus of this paper is the aggregation part of GCN. While the experiment results empirically demonstrate that our analysis seems to hold with non-linearity, more formal investigation for GCN with non-linearity is valuable. Our analysis in Theorem 2 assumes the independence between features, which limits the generality of the analysis and we would like to conduct further instigation to more general case. We provide theoretical understanding on GCN's performance based on CSBM, which stands for a type of graphs

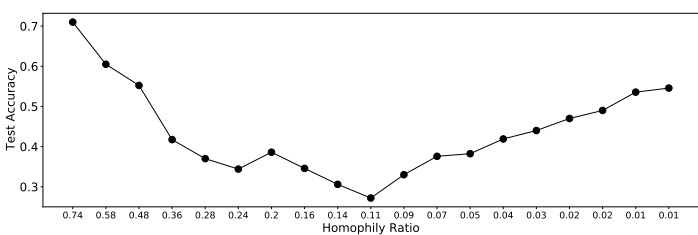 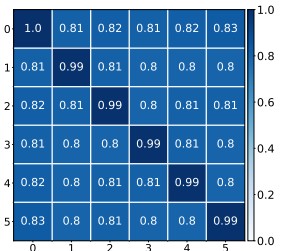

(a) Performance of GCN on synthetic graphs from `Citeseer`. Graphs are generated following *all* neighborhood distribution.

(b) Cross-class neighborhood similarity for the graph generated from `Citeseer` following *all* with homophily ratio 0.01.

Figure 19: `Citeseer`: *all* neighborhood distribution pattern

attracting increasing attention in the research community. However, it is not ideal for modeling sparse graphs, which are commonly observed in the real-world. Hence, it is important to devote more efforts to analyzing more general graphs. We believe our results established a solid initial study for further investigation. Finally, our current theoretical analysis majorly focuses on the GCN model; we hope to extend this analysis in the future to more general message-passing neural networks.

## J  BROADER IMPACT

Graph neural networks (GNNs) are a prominent architecture for modeling and understanding graph-structured data in a variety of practical applications. Most GNNs have a natural inductive bias towards leveraging graph neighborhood information to make inferences, which can exacerbate unfair or biased outcomes during inference, especially when such neighborhoods are formed according to inherently biased upstream processes, e.g. rich-get-richer phenomena and other disparities in the opportunities to "connect" to other nodes: For example, older papers garner more citations than newer ones, and are hence likely to have a higher in-degree in citation networks and hence benefit more from neighborhood information; similar analogs can be drawn for more established webpages attracting more attention in search results. Professional networking ("ability to connect") may be easier for those individuals (nodes) who are at top-tier, well-funded universities compared to those who are not. Such factors influence network formation, sparsity, and thus GNN inference quality simply due to network topology (Tang et al., 2020b). Given these acknowledged issues, GNNs are still used in applications including ranking (Sankar et al., 2021), recommendation(Jain and Molino), engagement prediction (Tang et al., 2020a), traffic modeling(Jiang and Luo, 2021), search and discovery (Ying et al., 2018) and more, and when unchecked, suffer traditional machine learning unfairness issues (Dai and Wang, 2021).

Despite these practical impacts, the prominent notion in prior literature in this space has been that such methods are inapplicable or perform poorly on heterophilous graphs, and this may have mitigated practitioners' interests in applying such methods for ML problems in those domains conventionally considered heterophily-dominant (e.g. dating networks). Our work shows that this notion is misleading, and that heterophily and homophily are not themselves responsible for good or bad inference performance. We anticipate this finding to be helpful in furthering research into the capacity of GNN models to work in diverse data settings, and emphasize that our work provides an understanding, rather than a new methodology or approach, and thus do not anticipate negative broader impacts from our findings.

## ACKNOWLEDGEMENTS

This research is supported by the National Science Foundation (NSF) under grant numbers IIS1714741, CNS1815636, IIS1845081, IIS1907704, IIS1928278, IIS1955285, IOS2107215, and

IOS2035472, the Army Research Office (ARO) under grant number W911NF-21-1-0198, the Home Depot, Cisco Systems Inc and Snap Inc.

