# OpenReview forum: "Is Homophily a Necessity for Graph Neural Networks?"
_ICLR.cc/2022/Conference — ICLR 2022 Poster_

### Official Review · Reviewer_MC2o · 2021-10-30

**Correctness:** 3
**Technical Novelty And Significance:** 3
**Empirical Novelty And Significance:** 2
**Recommendation:** 6
**Confidence:** 4

**Main Review:**

The paper aims to provide a more careful assessment about the role of label-homophily for graph neural network. Given the research activity  in this area recently, I think this is an interesting topic. As the authors outline, there appears to be indeed some "folklore knowledge" about label-homophily present in the literature which is probably not entirely accurate. Establishing a more precise understanding on when CGNs can work well, and when not is indeed useful.

However, what I find problematic is that the authors claim that the current understanding in the literature is that homophily would be a necessity for CGN to perform well (i.e., a necessary condition for CGNs to work). This feels to much like a straw man argument for me; I think in most cases the argument is that heterophily can be problematic for CGNs -- and indeed the authors confirm this as well: "bad homophily" does indeed exist, as they say. The claim that there was some kind of consensus that CGNs can never perform well in label-heterophilic graphs is too far reaching to me.
The authors have to be very careful here to not throw out the baby with the bathwater. There is merit in what the authors do: clarifying when CGNs can perform well (even under heterophily); however, that does not mean that there are no issues with heterophilic graphs in general. I strongly suggest the authors reconsider their phrasing in certain areas.

In a similar vein, I believe the paper would benefit from a more precise treatment of certain aspects of homophily as well. Basically, the authors consider a homophily definition at the level of node-labels; they also assume that there is a tight correlation between the node labels and the underlying node features; so a label-homophily would translate into a feature homophily (or heterophily) as well. I think some more in depth discussion in this context about the relation between the correlation of features of neighbors and labels would be helpful.

At a high level, CGNs act by aggregating (averaging) the features of the node neighbors. If those features in the neighborhood are correlated (or indeed anti-correlated) to the target node label, then CGNs will be able to pick out the correct relation. This is true for fully homophilic graphs -- in which there is a strong positive correlation between the neighborhood features and the true node label -- and, e.g., for the bipartite example the authors present, in which case there is a strong anti-correlation (which works for prediction as well). However, extending the bipartite examples to multiple classes is already much more complicated: it may well happen that there is no clear label-to-feature correspondence any more that can be exploited. Some aspects of this are already alluded to in the discussion on the embeddings -- but I believe need to be more clearly articulated.

Some further technical comments:

The CGN architecture described by the authors has no self-embedding component; this is an important point as it leads to "no mixing" behavior of the features / labels the authors describe for the toy example. Most convolutional architectures of GNNs have however at least some diagonal weights / self-embeddings -- which will inevitably lead to a stronger mixing between features of nodes of different classes. Again it is the correlation structure between node-labels (across the graph) and node-features (features to labels and features across the graph) that will come into play here and self-loops may help or hinder here.

While the probabilistic analysis provided with the SBMs is clearly better than simply assuming constant features; performing the analysis in expectation is in this case basically the same as the deterministic construction discussed at the end of page 3, augmented with a bound on the deviation bound (which is only available here because of the bounded feature assumption it seems)

There are at least some problematic aspects with this analysis:
1) the SBM and alike models are not a good model for sparse graphs, as they exhibit a graphon like structure and lead to dense graphs -- which is problematic as we often consider sparse graphs in practise; this limitation needs to be at least discussed I think.

2) the assumption that the every node with the same class label has the same feature distribution (thus providing a perfect probabilistic coupling between labels and features) and the same neighbor label distribution and that each feature is indeed independent seems very unrealistic. This would lead basically to a problem in which a single inspection of the neighborhood suffices -- there is (almost) no graph structure necessary here really.

3) Dropping the nonlinearity, while done in practice, is clearly another strong simplification which would merit some more discussion I think.

Finally, the numerical experiments paint a much richer picture than the theoretical analysis and indeed the introduction of the paper. Some of these aspects could also be discussed in the theoretical parts (neighborhood similarity and correlation structure) -- which would lead to a more balanced picture. As it stands the introduction and theory parts make the paper feel a bit one-sides (to exaggerate: "heterophily is not a problem"); whereas in the end the conclusion is much more nuanced. I think such a more nuanced appraisal of heterophily/homophily is useful throughout.

Minor comments
p.2 last line -- X is never defined, instead H and H' are used as input/output. Please correct.



**Summary Of The Paper:**

The paper discusses the relationships between the performance of GNNs for semi-supervised node-classification taaks and label-homophily/heterophily in the underlying graph structure.
The authors show that the classic CGN architecture can perform well on certain label heterophilic graphs -- which they argue is disputed heavily in the literature -- and provide a theoretical discussion on when a good performance for CGN on such heterophilic graphs is possible.

**Summary Of The Review:**

Overall I see the paper slightly above the line, provided some of the above points are addressed (I think this is possible).

The paper should be improved with respect to the following points (see above for more detailed suggestions and comments)
a) provide a more nuanced appraisal of literature and the role of homophily/heterophiliy
b) discuss the roles of the correlation structure between node labels and features more clearly
c) discuss the role of self-loops/ego embedding in mixing information from different classes
d) provide a more rigorous discussion on the assumptions and shortcomings of the theoretical analysis and the SBM model.

---

> ### Author Response · Authors · 2021-11-23
> **Response to Reviewer MC2o--Part 1**
>
> We thank the reviewer for providing insightful and constructive comments. We also appreciate the reviewer's recognization of the novelty and contribution of our work. We address the reviewer's comments/concerns as follows.
>
> ---
>
> Q1: However, what I find problematic is that the authors claim that the current understanding in the literature is that homophily would be a necessity for CGN to perform well (i.e., a necessary condition for CGNs to work). This feels to much like a straw man argument for me; I think in most cases the argument is that heterophily can be problematic for CGNs -- and indeed the authors confirm this as well: "bad homophily" does indeed exist, as they say. The claim that there was some kind of consensus that CGNs can never perform well in label-heterophilic graphs is too far reaching to me. The authors have to be very careful here to not throw out the baby with the bathwater. There is merit in what the authors do: clarifying when CGNs can perform well (even under heterophily); however, that does not mean that there are no issues with heterophilic graphs in general. I strongly suggest the authors reconsider their phrasing in certain areas.
>
> A1: We have adjusted the contents in the introduction. More specifically, we change "existing literature posits that strong homophily of the underlying graph is a necessity for GNNs to achieve good performance on SSNC" to "several existing literature posits that many GNNs implicitly assume strong homophily and homophily is critical for GNNs to achieve strong performance on SSNC". The latter is widely spreading in the community and commonly claimed in existing literature [1,2,3,4,5]. Several other medications are also made in the abstraction and introduction to make our claim more accurate. These modifications are highlighted in blue.
>
> [1] Large Scale Learning on Non-Homophilous Graphs: New Benchmarks and Strong Simple Methods. NeurIPs 2021
>
> [2] Graph Neural Networks with Heterophily, AAAI 2021.
>
> [3] Beyond Homophily in Graph Neural Networks: Current Limitations and Effective Designs. NeurIPs 2020.
>
> [4] Beyond Low-Pass Filters: Adaptive Feature Propagation on Graphs. ECML-PKDD, 2021
>
> [5] Adaptive Universal Generalized PageRank Graph Neural Network. ICLR, 2021.
>
> ---

---

> > ### Author Response · Authors · 2021-11-23
> > **Response to Reviewer MC2o--Part 2**
> >
> > Q2: In a similar vein, I believe the paper would benefit from a more precise treatment of certain aspects of homophily as well. Basically, the authors consider a homophily definition at the level of node-labels; they also assume that there is a tight correlation between the node labels and the underlying node features; so a label-homophily would translate into a feature homophily (or heterophily) as well. I think some more in depth discussion in this context about the relation between the correlation of features of neighbors and labels would be helpful.
> >
> > At a high level, CGNs act by aggregating (averaging) the features of the node neighbors. If those features in the neighborhood are correlated (or indeed anti-correlated) to the target node label, then CGNs will be able to pick out the correct relation. This is true for fully homophilic graphs -- in which there is a strong positive correlation between the neighborhood features and the true node label -- and, e.g., for the bipartite example the authors present, in which case there is a strong anti-correlation (which works for prediction as well). However, extending the bipartite examples to multiple classes is already much more complicated: it may well happen that there is no clear label-to-feature correspondence any more that can be exploited. Some aspects of this are already alluded to in the discussion on the embeddings -- but I believe need to be more clearly articulated.
> >
> > A2: Thanks for the insightful comments and discussion. We find these comments very helpful, which provides a new perspective to understand our paper. Indeed, the correlation between node features (before or after GCN) and labels are the key for classification performance. In this paper, we generally assume that the original features have some kind of correlation with the labels. However, we do not need to make the assumption that "features in the neighborhood are correlated (or indeed anti-correlated) to the target node label", which is not required for our analysis. For example, in the binary CSBM in Section 3.2, we assume the features of nodes with the same label are sampled from a label-specific Gaussian distribution. But we do not assume the correlation between the features of one class to the other class. The argument " If those features in the neighborhood are correlated (or indeed anti-correlated) to the target node label, then CGNs will be able to pick out the correct relation." is not entirely precise. Indeed the correlation between the output features from GCN and the target node label is important for the classification performance. However, such correlation is not due to the correlation between the neighborhood features and the target node label. Instead, it is from the neighborhood distribution. The key idea in our analysis is that when same-labeled nodes follow the same neighborhood distribution, the aggregation process in GCN guarantees the correlation between the output feature from GCN and the target label. If same-labeled nodes do not follow the same neighborhood distribution, such correlation will be broken, i.e, nodes from the same label may be embedded in different areas in the output feature space. We emphasize that the correlation between the output features from GCN and the node label is due to the assumption of having the same neighborhood distribution for same-label nodes. It does not require the correlation between the center node label and the features of neighboring nodes.
> >
> > Note that our analysis and results are not limited to binary cases (or bipartite cases). They can be generalized to cases with multiple classes. Specifically, our analysis in Theorem 2 can be extended to multiple classes. We include the extended analysis in Appendix F. The empirical results in Section 3.2 are also based on cases with multiple classes.
> >
> > ---
> >
> > Q3: The CGN architecture described by the authors has no self-embedding component; this is an important point as it leads to "no mixing" behavior of the features / labels the authors describe for the toy example. Most convolutional architectures of GNNs have however at least some diagonal weights / self-embeddings -- which will inevitably lead to a stronger mixing between features of nodes of different classes. Again it is the correlation structure between node-labels (across the graph) and node-features (features to labels and features across the graph) that will come into play here and self-loops may help or hinder here.

---

> > > ### Author Response · Authors · 2021-11-23
> > > **Response to Reviewer MC2o--Part 3**
> > >
> > > A3: Thanks for the insightful comments. We discuss how ego-embedding may affect our analysis. For GCN, the ego-embedding is usually included by weighted average, i.e, the features of the target node are treated equally as all its neighbors. Intuitively, during the aggregation, the importance of the features of a target node $i$ is only $1/(deg(i)+1)$, where $deg(i)$ denotes the degree of node $i$. Hence, the ego-embedding will not affect the analysis very significantly for those nodes with a large degree. It will impact low-degree nodes more. For methods such as GraphSage, which combines ego-embedding and neighborhood embedding through concatenation (and transformation), further analyses needed to be extended from our current results. In this case, ego-embedding is treated equally as the entire neighborhood information. The neighborhood information is typically aggregated by feature average, which is what we discussed in this paper. Hence, our analysis in this paper can be directly utilized for analyzing methods such as GraphSage. As mentioned by the reviewer, combining ego-embedding and neighborhood embedding could be either good or bad. We take different heterophilous kinds of graphs for discussion. For graphs with ``bad heterophily'' (for example, nodes from the same label follow different neighborhood distribution patterns), the neighborhood information is generally not helpful for classification. Then, in this case, methods such as GraphSage may achieve stronger performance as it at least has its own ego-embedding playing an important role in the final features while in the GCN-like method the ego-embedding information is buried in the neighborhood information. We think this is definitely worth more discussion and investigation, which we leave for future work.
> > >
> > > ---
> > > Q4: While the probabilistic analysis provided with the SBMs is clearly better than simply assuming constant features; performing the analysis in expectation is in this case basically the same as the deterministic construction discussed at the end of page 3, augmented with a bound on the deviation bound (which is only available here because of the bounded feature assumption it seems)
> > >
> > > A4: We agree that performing the analysis for CSBM in expectation is similar to the deterministic construction discussed at the end of page 3. However, our analysis is not solely based on expectations. Specifically, Theorem 2 compares the misclassification probability for any node $i$ before and after the GCN operation. This analysis involves the distribution of the output features (not only its expectation). More specifically, the expectations of the two classes are utilized to determine the distance between the two classes (inter-class distance). The larger the inter-class distance, the easier the classification is. The standard deviation of the distribution of the output features determines the intra-class distance. The smaller the standard deviation is, the closer the embeddings for same-labeled nodes are (this can roughly be understood as a stronger correlation). In fact, in the proof of Theorem 2, we demonstrate that, after GCN operation, the inter-class distance is reduced by a factor of $|p-q|/(p+q)$ and the standard deviation for the feature distribution (for features output from GCN) is reduced by a factor of $\sqrt{deg(i)}$. Roughly speaking, when the effect of reducing standard deviation is more significant than the reduction of inter-class distance, the GCN model can help reduce the misclassification rate. This is how we reached the threshold in Theorem 2. Detailed proof can be found in Appendix B. In summary, the probabilistic analysis with CSBM provided more precise and formal characterization for GCN's performance than those in "the deterministic construction discussed at the end of page 3".
> > >
> > > ---
> > >
> > > Q5: the SBM and alike models are not a good model for sparse graphs, as they exhibit a graphon-like structure and lead to dense graphs -- which is problematic as we often consider sparse graphs in practice; this limitation needs to be at least discussed I think.
> > >
> > > A5: We adopt the SBM model to make the analysis more feasible since many of its key properties can be conveniently controlled. Also, it is widely adopted for investigating algorithms on graphs. We agree that SBM model is not very ideal for modeling sparse graphs. We discuss this limitation in the Section of the Conclusion. We also agree that it is important to perform analysis for more general graphs, which will be more complicated and a bit out of the scope of this paper as it is only an initial piece of this research direction. We generally believe our analysis provides a solid step towards deeper analysis on more general graphs.

---

> > > > ### Author Response · Authors · 2021-11-23
> > > > **Response to Reviewer MC2o--Part 4**
> > > >
> > > >
> > > > ---
> > > > Q6: the assumption that every node with the same class label has the same feature distribution (thus providing a perfect probabilistic coupling between labels and features) and the same neighbor label distribution and that each feature is indeed independent seems very unrealistic. This would lead basically to a problem in which a single inspection of the neighborhood suffices -- there is (almost) no graph structure necessary here really.
> > > >
> > > > A6: Note that our goal in this paper is to characterize the graphs where GCNs can perform well and these assumptions help such analysis. We provide the rationale for each of the motioned three assumptions.
> > > >
> > > > 1) The assumption that "every node with the same class label has the same feature distribution" is commonly adopted in GCN/graph literature [1,2,3]. These distributions we assumed are conditional probability distributions, which can be formulated as  $p({\bf x}|y_c)$ for a given label $y_c$, where ${\bf x}$ denotes features (for those nodes with label $y_c$). It assumes that there exists some kind of correlation between the label and features, which is typically required for learning classification models. For example, the generative models for classification aims to estimate such conditional probability distribution $p({\bf x}|y_c)$ for all $y_c$ from the given data.
> > > >
> > > > 2) The assumption for neighborhood distribution is directly assumed to characterize those graphs (no matter homophily or heterophily) where the GCNs can perform reasonably well, i.e, this is part of our contribution.
> > > >
> > > > 3) The assumption that "each feature is indeed independent" is also adopted in some other machine learning methods such as the Naive Bayes classifier. We agree that this assumption is relatively unrealistic. We add a discussion on this limitation in the Section of the Conclusion.
> > > >
> > > > Note that in the classification task, we are only given the features but not the labels as we are predicting the labels. Hence, a single inspection of the neighborhood (if the reviewer means to count the labels of neighboring nodes) is not enough for classification. We need to aggregate the features and combine them for classification, which is what GCN does. The neighborhood information is an important kind of graph structure information. The GCN model indeed captures such neighborhood information.
> > > >
> > > > [1] Graph Convolution for Semi-Supervised Classification: Improved Linear Separability and Out-of-Distribution Generalization. ICML 2021.
> > > >
> > > > [2] Contextual Stochastic Block Models. NeruIPs 2018.
> > > >
> > > > [3] Adaptive Universal Generalized Pagerank Graph Neural Network. ICLR 2021
> > > >
> > > > ---
> > > >
> > > > Q7: Dropping the nonlinearity, while done in practice, is clearly another strong simplification that would merit some more discussion I think.
> > > >
> > > > A7:   In this paper, our analysis mainly focuses on understanding the aggregation part of the GCN model. Hence, we follow some previous works to drop the non-linearity. Furthermore, the empirical results demonstrate that our analysis is likely to be valid even with non-linearity. Nonetheless, we see the dropping of the non-linearity as a limitation of our work and discuss this limitation in the Section of the Conclusion.
> > > >
> > > > ---
> > > >
> > > > Q8: Finally, the numerical experiments paint a much richer picture than the theoretical analysis and indeed the introduction of the paper. Some of these aspects could also be discussed in the theoretical parts (neighborhood similarity and correlation structure) -- which would lead to a more balanced picture. As it stands the introduction and theory parts make the paper feel a bit one-sided (to exaggerate: "heterophily is not a problem"); whereas in the end, the conclusion is much more nuanced. I think such a more nuanced appraisal of heterophily/homophily is useful throughout.
> > > >
> > > > A8: Thanks for the suggestion. We have adjusted the introduction to make it more balance. Specifically, we now clearly state that there exist both "good" and "bad" heterophily in the introduction. We also correspondingly adjusted the theory part to make it more balanced. The modifications are highlighted in blue.
> > > >
> > > > ---
> > > >
> > > > Q9: Minor comments p.2 last line -- X is never defined, instead H and H' are used as input/output. Please correct.
> > > >
> > > > A9: Thanks for pointing this out. We have corrected this typo.
> > > >
> > > > ---
> > > > We believe that we have responded to and addressed all your concerns with our revisions. Please let us know in case there are outstanding concerns, and if so, we will be happy to respond.

---

### Official Review · Reviewer_WXGg · 2021-10-31

**Correctness:** 3
**Technical Novelty And Significance:** 3
**Empirical Novelty And Significance:** 2
**Recommendation:** 6
**Confidence:** 4

**Main Review:**

This work proposes a novel perspective on learning node representations under various homophily. Considering how recent GNN papers on heterophily treat GCNs (as the weakest baseline), we can say that empirical results by this paper are groundbreaking. However, this paper has several flaws which should be fixed/justified before publication.

First, the Table 1 results are not convincing. We now agree that GCNs can learn good representations under some conditions on neighbor label distributions. However, this cannot justify how GCNs outperform other models. Does it not hold for other specialized architectures (H2GCN, CPGNN, GRRGNN)? Or, how about simple other baselines such as GraphSAGE and GAT?

Related to the first point, is there any reason for GCNs’ supremacy other than hyperparameter tuning? I have read Appendix D.4, and it is a standard procedure with reasonable computational budgets. Does it mean that other related works were doing something wrong? What is the magic here?

Second, the edge addition algorithm does not control the degree distribution; it monotonically increases the degree of nodes. In Theorem 1, the probability is bounded with singular value, feature dimension, and degree. However, the edge addition algorithm adds edges, and the average degree always increases; thus, whole experiments in Figure 3 do not control the key variate. Does it contribute to getting over-optimistic results for a high degree (or low homophily) regime?

Third, the two-class graph results seem to be insufficient for the main claim. Theorem 2 is about the relation between decision boundary and degree, $p$, and $q$, and this is just one example of a graph with distinguishable neighbor label patterns. Can we say that Theorem 2 *theoretically supports* the paper’s argument? This can work just because predicting one can be reducible to predicting the other in a two-class problem, might not because of the distinguishability.

Fourth, the degree distribution $\mathcal{D}_c$ for edge addition is clearly distinguishable to each other, but only one specific instance has been experimented with (i.e., having neighbors of two other classes other than itself). Is there any specific reason to experiment with one pattern across all datasets? Can we confirm that these empirical results generalize to various neighbor label distributions? For extreme cases, how does the model perform if the node connects to a single different label, or all labels except for itself (with the different number of classes)?


**Summary Of The Paper:**

This paper revisits the common belief in prior works, “GCNs require strong homophily assumption.” The paper claims that even under low homophily, if the nodes of each label have distinguishable neighbor label distributions, GCNs can learn distinguishable node representations. To prove this claim, the authors (1) show that GCNs outperform representative baselines on two heterophilous graphs, (2) theoretically analyze the relation between homophily and decision boundary on CSBM with two classes, (3) conduct experiments on synthetic benchmarks created by distinct/random edge addition, and (4) analyze real-world benchmarks qualitatively and quantitatively with the lens of (3).


**Summary Of The Review:**

The authors discuss the interesting question (in the title) and their answers in theoretical and empirical ways. The paper has merits but also has the following weaknesses:

- Table 1 and the following description cannot justify how GCNs outperform other models.
- The edge addition algorithm does not control the degrees, which is the core control variate for the experiments.
- The two-class graph results seem to be insufficient for the main claim.
- Only one specific instance of the degree distribution $\mathcal{D}_c$ for edge addition is used for the experiments.

---

> ### Author Response · Authors · 2021-11-23
> **Response to Reviewer WXGg--Part 1**
>
> We appreciate the reviewer's recognition of the novelty and contribution of this paper. To address the reviewer's concerns/comments, we provide the following detailed responses.
>
> ---
> Q1:  First, the Table 1 results are not convincing. We now agree that GCNs can learn good representations under some conditions on neighbor label distributions. However, this cannot justify how GCNs outperform other models. Does it not hold for other specialized architectures (H2GCN, CPGNN, GRRGNN)? Or, how about simple other baselines such as GraphSAGE and GAT?
>
> A1: The main goal of this paper is not to compare GCNs with other architectures. Instead, we aim to demonstrate that GCNs can achieve reasonable performance for some heterophilous graphs under certain conditions. Thus, the analysis in this paper mainly focuses on GCNs. The analysis we developed for the GCN model does not exactly hold for other specialized architectures (H2GCN, CPGNN, GPRGNN) or other simple models such as GraphSage and GAT, as they usually introduce different designs. For example, H2GNN introduces several key designs which make the model more complicated than simple aggregation. In GAT, the attention mechanism is utilized to determine the importance of different neighboring nodes. These designs make them capture different information from GCNs, which leads to different performance compared with GCNs (either better or worse than GCNs). Compared with GCNs, these designs may lead to either better or worse performance depending on the datasets and tasks. On the Chameleon and Squirrel, these other models potentially failed to capture as much useful information as GCNs and thus their performance is not as good as GCNs. We believe deeper investigations/analyses are needed to gain more understandings of these frameworks. However, such analysis requires a significant amount of additional effort and is out of the scope of this paper. We leave it for future work.
>
> ---
> Q2: Related to the first point, is there any reason for GCNs’ supremacy other than hyperparameter tuning? I have read Appendix D.4, and it is a standard procedure with reasonable computational budgets. Does it mean that other related works were doing something wrong? What is the magic here?
>
> A2: We clarify that differences in hyper-parameters may lead to the discrepancy of performances reported in our paper and those in the other related papers such as H2GNN. For example, the hyper-parameter of weight_decay has a significant impact on the GCN's performance. In the Chameleon dataset, when we adopt $5e-04$ as weight_decay  rather than $5e-07$  (the one we currently use), GCN's performance drops from about 68% to around 60% (close to the one reported in H2GNN). Similarly, GCN's performance on Squirrel drops to around 36% when adopting $5e-04$ instead of $5e-07$ as weight_decay. Hence, the search spaces of the hyper-parameters may be the potential reason why our results are different from other related papers.  For reproducing the results in Table 1, we have provided the code with detailed hyper-parameter settings in the following link (https://drive.google.com/file/d/17daB5ZvHEEJgVJawGTb6V-V1udLY445L/view?usp=sharing). We hope this could help clarify the confusion on results in Table 1.

---

> > ### Author Response · Authors · 2021-11-23
> > **Response to Reviewer WXGg--Part 2**
> >
> > Q3: Second, the edge addition algorithm does not control the degree distribution; it monotonically increases the degree of nodes. In Theorem 1, the probability is bounded with singular value, feature dimension, and degree. However, the edge addition algorithm adds edges, and the average degree always increases; thus, whole experiments in Figure 3 do not control the key variate. Does it contribute to getting over-optimistic results for a high degree (or low homophily) regime?
> >
> > A3: Note that the experiment results in Section 3 are not meant to demonstrate Theorem 1. Our goal is to show that GCNs are actually able to achieve reasonable performance for both homophilous **and heterophilous** graphs if they follow certain assumptions as discussed in the two Observations. Furthermore, we also aim to demonstrate that there are both "good" heterophily and "bad" heterophily; GCNs are able to achieve strong performance for "good" heterophily but not for "bad" heterophily. Our results are presented to clearly demonstrate these claims. These claims themselves are not related to node degree since we do not aim to tell whether homophily or heterophily is better (in which case, we need to ensure a fair comparison by controlling the degree). No matter how the degree changes, our generated extremely heterophilous graphs (potentially with large degrees) are still heterophilous. So, if GCNs achieve good performance on these heterophilous graphs (specifically see the few right points in the black line in Figure 3), we believe it reasonably demonstrates our first claim that GCNs are able to achieve reasonable performance for some heterophilous graphs. Also, as discussed in Section 3.2.2, if we observe Figure 3 vertically, i.e, by comparing various graphs with the same homophily ratio, we can find the existence of "good" and "bad" heterophily over multiple graphs with the same average degree (all these graphs have the same number of edges added by definition). For example, on Cora, GCN's performances for graphs with homophily ratio $h=0.25$ varies a lot, ranging from very high (nearly 90\%) accuracy to very low (nearly 40\%) accuracy -- this further demonstrates the existence of  "good" and "bad" heterophily.
> >
> > We agree that the degree is an important factor affecting GCN's performance as the degree impacts the intra-class distance. We also discussed this in Theorem 2 for the binary CSBM, which is further extended to multiple classes cases in Appendix F in the revision. Hence, to decouple the impact from the degree, we further conduct additional experiments with a controlled average degree in Appendix H.1. In this set of experiments, when generating new graphs, we control the number of edges to be the same as the corresponding original graphs. Thus, the average degree of the generated graphs will be the same as the original graph since the number of nodes for these graphs is the same. Specifically, while adding $K$ inter-class edges according to Algorithm 1, we correspondingly remove $K$ intra-class edges. We can then generate different graphs by varying the value of $K$. Note that $K$ is at most as large as the total number of intra-class edges in the original graph (in which case the generated graph will have a homophily ratio $0$). We demonstrate the GCN's performance on these graphs with the same degree in Figure 12 (Figure 12(a) for Cora; Figure 12(b) for Citeseer). Clearly, we can still observe the $V$-shape curve for both datasets. However, even when the homophily ratio is $0$, GCN cannot achieve perfect performance, which is different from what we observed in Figure 7 in Appendix C.3. This indeed demonstrates the impact of the degree to GCN's performance -- namely, that GCN is benefited by higher degrees -- and is consistent with the discussion in Observation 2, as well as Theorem 2 and the text following it.

---

> > > ### Author Response · Authors · 2021-11-23
> > > **Response to Reviewer WXGg--Part 3**
> > >
> > > Q4: Third, the two-class graph results seem to be insufficient for the main claim. Theorem 2 is about the relation between decision boundary and degree, $p$ and $q$, and this is just one example of a graph with distinguishable neighbor label patterns. Can we say that Theorem 2 theoretically supports the paper’s argument? This can work just because predicting one can be reducible to predicting the other in a two-class problem, might not because of the distinguishability.
> > >
> > > A4: The two-class CSBM is the model we adopt to support our claim since we can conveniently control the distinguishability of the neighborhood distributions for the two classes through $p$ and $q$. Theorem 2 can support the paper's argument for graphs generated from the introduced binary CSBM. Note that Theorem 2 demonstrates when GCN can improve the linear separability based on the distinguishability of neighborhood distributions and the node degree (check detailed discussions in the text following Theorem 2). The GCN works not "because predicting one can be reducible to predicting the other in a two-class problem". There are binary classification cases where GCN does not work. The GCN can work only when the distinguishability between the distributions is ensured. For example, as we discussed in the paragraph following Theorem 2, for the CSBM, when $p=q$, the neighborhood distributions are completely indistinguishable. In this case, the GCN cannot help improve the separability at all. This clearly demonstrates that being a "two-class problem" is not the reason why it works and the distinguishability is important. Furthermore, the analysis can be extended to CSBM with multiple classes (we provide a proof sketch in Appendix F). We note that CSBM stands for a special type of graph model that is widely adopted in graph research. More efforts are needed to extend the analysis for more general graphs. We believe that our work provides a solid first step towards deeper analysis for more general cases.

---

> > > > ### Author Response · Authors · 2021-11-23
> > > > **Response to Reviewer WXGg--Part 4**
> > > >
> > > > ---
> > > > Q5: Fourth, the degree distribution for edge addition is clearly distinguishable to each other, but only one specific instance has been experimented with (i.e., having neighbors of two other classes other than itself). Is there any specific reason to experiment with one pattern across all datasets? Can we confirm that these empirical results generalize to various neighbor label distributions? For extreme cases, how does the model perform if the node connects to a single different label, or all labels except for itself (with the different number of classes)?
> > > >
> > > > A5:  Our analysis is for general neighborhood distributions and it is not limited to the patterns we adopt in the paper. We choose such a circulant pattern for both datasets for the convenience of description. Note that it is impractical to enumerate all possible neighborhood distributions. To demonstrate that the analysis holds for other neighborhood distributions patterns, we provide more experiments with other neighborhood distribution patterns. These patterns include two additional neighborhood distribution patterns for both datasets and also the two extreme cases mentioned by the reviewer. A detailed description of the experiments and the results can be found in Appendix H.2 of the revised paper.
> > > >
> > > > In Section H.2.1, we provide the results for the two additional sets of neighborhood distribution patterns ( check the detailed neighborhood distributions in Section H.2.1). For both Cora and Citeseer, the behaviors for these graphs generated following these new neighborhood distribution patterns are similar to those we observed in Figure 3 (the black curves). More specifically, we can clearly observe the $V$-shape curve under these new settings. Please check the performance of GCNs for graphs generated with these distributions in Section H.2.1 (Figure 13 and Figure 14 for Cora; Figure 15 and Figure 16 for Citeseer).
> > > >
> > > > In Section H.2.2, we provide the results for the two extreme neighborhood patterns. We provide detailed information about these neighborhood distributions and detailed analyses of GCN's performance in Section H.2.2. We provide a concise summary of the observations and analysis here. In the extreme case of  "node connects to a single different label", the neighborhood distributions for different labels are highly distinguishable (no overlap between neighborhood distributions of different labels). Hence, the performance of GCNs is similar to those in Figure 3 (the black curves) and those in Figure 7 in Appendix C.3. Please check the performance of GCNs for graphs generated in this case in Figure 17 (for Cora) and Figure 19 (for Citeseer).  In the extreme case of  "node connects to all labels except for itself", we can still observe the $V$-shape curves as in Figure 3. However, we cannot achieve perfect performance even when we add an extremely large number of edges to the graphs (which is possible with other more distinguishable distribution patterns such as the one in Figure 7 in Appendix C.3). This is because, in this case, the neighborhood distributions of different labels are not easily distinguishable (much harder compared with the other extreme case). More specifically, any pair of classes share "$C-2$ labels" in their neighborhood distribution patterns with $C$ denoting the total number of labels (please check the distributions provided in Appendix H.2.2 for more details). In this case, the distinguishability of neighborhood distributions for different labels is not high enough (or the "heterophily" is not such "good"), and thus the performance of GCN is limited. This observation further demonstrates our key argument that the distinguishability of the distributions for different labels is important for GCN's performance. More detailed analysis and discussions can be found in H.2.2. The performance of GCNs for graphs generated in this case can be found in Figure 18 (for Cora) and in Figure 20 (for Citeseer).
> > > >
> > > > ---
> > > >
> > > > We believe that we have carefully responded to and addressed all your concerns with our revisions — in light of this, we hope you consider raising the score. Please let us know in case there are outstanding concerns, and if so, we will be happy to respond.

---

### Official Review · Reviewer_6L9A · 2021-11-03

**Correctness:** 2
**Technical Novelty And Significance:** 3
**Empirical Novelty And Significance:** 3
**Recommendation:** 6
**Confidence:** 4

**Main Review:**

Strength:
1.	It provides a deeper insights into the heterophily problem and help us understand the GCNs’ performance.
2.	It also designs a new cross-class neighborhood similarity metric to help explain the performance of GCN on various graphs (although it is not perfect).
3.	The writing is generally clear.


Weakness and Questions:
1.	The analysis is only limited to GCNs, while the paper title is too general (GNNs). Not very
2.	One concern is that the performance of GCN on Chameleon and Squirrel (Table1) differs a lot from the one reported in other papers (e.g. Geom-GCN (Pei et al. 2020), H2GNN (Zhu et al. 2020)). It seems the settings are the same as Pei et al. 2020, why is the result on these two datasets so different (2 to 3 times different)? Generally, I do not think the hyperprameter tunning should impact so much. Could the authors explain more details about it? In fact I also run some experiments on those datasets before and I cannot get those high numbers either.
3.	The cross-class neighborhood similarity metric is intuitive and a good idea. However, it lacks of a direct theoretic connection to GCNs’ performance. Same as the heterophily metric, I do not think it can completely decide the GCNs’ performance, because the node feature distribution is also important here.  In fact, the assumptions in Theorem 1 are quite strong. If the nodes with the same label are sampled from the same feature distribution, it means generally MLP can also have a good performance. When this assumption does not meet, the analysis will become very complex. That is why I think Figure5 may be not enough to explain everything (but it is still interesting to see this empirical result).
4.	Although Theorem 1 seems correct to me, I have a question here. Assume we have a separate node with 0 neighbors, that means the upper bound here is 0. It is obviously not true. So, how to explain this exception?


**Summary Of The Paper:**

The paper revisited the performance of GCN on graph with heterophily and provide negative evidence that heterophily does not always result in the poor performance of GCN, which contradicts with the assumptions/observations of many previous papers. They demonstrated that the GCN embeddings are still label-distinguishable on a special type of graphs with assumptions that the nodes with same labels have the same node feature distribution as well as the same neighborhood label distributions. They theoretically analyzed the case of CSBM model with two classes, and also empirically investigated on synthetic/real-world graphs with multiple classes. The conclusion is that GCNs can achieve good performance on heterophilous graphs under certain conditions.

**Summary Of The Review:**

The paper provides a new perspective of heterophily and GCNs, but there remains some concerns both in the theoretic part and in the experiments.

---

> ### Author Response · Authors · 2021-11-23
> **Response to Reviewer 6L9A--Part 1**
>
> We thank the reviewer for their efforts and time in reviewing our paper. Also, we appreciate the reviewer's perception of our key novelty and contributions. We provided detailed responses to the reviewer's comments/questions as follows.
>
> ---
>
> Q1: The analysis is only limited to GCNs, while the paper title is too general (GNNs). Not very
>
> A1: We note that although our analyses are mainly for GCNs, a similar line of analysis may be used for more general message-passing neural networks (MPNNs). For example, the analysis could be directly helpful for analyzing some other GNN methods such as GraphSage. Specifically, for GraphSage, based on the current analysis, we only need to further consider the neighborhood sampling process and the combination of ego and neighbor features. Such extensions are important and valuable. Although our work mainly focuses on GCNs for simplicity in analysis, it provides a solid and important backbone for investigating other GNN methods as well. Such analysis may be nontrivial for other MPNNs which adopt more complicated designs. Hence, we decide to follow the reviewer's suggestion to change the reference to Graph Neural Networks to Graph Convolutional Networks in our title to keep the paper’s title in line with its contributions. However, changing the title of the paper through Openreview is not allowed during the rebuttal. We will update it in later revisions (camera-ready if accepted; re-submissions if rejected).
>
> ---
>
> Q2: One concern is that the performance of GCN on Chameleon and Squirrel (Table1) differs a lot from the one reported in other papers (e.g. Geom-GCN (Pei et al. 2020), H2GNN (Zhu et al. 2020)). It seems the settings are the same as Pei et al. 2020, why is the result on these two datasets so different (2 to 3 times different)? Generally, I do not think the hyperprameter tunning should impact so much. Could the authors explain more details about it? In fact, I also run some experiments on those datasets before and I cannot get those high numbers either.
>
> A2: We indeed follow the settings in Geom-GCN (Pei et al. 2020).  Note that GCN's performances reported in the H2GNN paper are 36.89% (Squirrel) and 59.82% (Chameleon), which are already much higher than those reported in the Geom-GCN paper (23.96% and 28.18%, correspondingly). Our results are not extremely far away from those reported in the H2GNN paper (not 2-3 times). Differences in hyper-parameters can potentially lead to the discrepancy of performances reported in our paper and those in the H2GNN paper. For example, the hyper-parameter weight_decay can significantly affect the performance.  In the Chameleon dataset, when we adopt $5e-04$ as weight_decay than $5e-07$ (the one we currently use), GCN's performance drops from 68% to around 60%. Similarly, GCN's performance on Squirrel drops to around 36% when adopting $5e-04$ instead of $5e-07$ as weight_decay. To further clarify the confusion on the results in Table 1, we have provided the code with detailed hyper-parameter settings in the following link (https://drive.google.com/file/d/17daB5ZvHEEJgVJawGTb6V-V1udLY445L/view?usp=sharing) to reproduce the results. We hope this could help clarify the confusion on results in Table 1.

---

> > ### Author Response · Authors · 2021-11-23
> > **Response to Reviewer 6L9A--Part 2**
> >
> >  Q3: The cross-class neighborhood similarity metric is intuitive and a good idea. However, it lacks of a direct theoretic connection to GCNs’ performance. Same as the heterophily metric, I do not think it can completely decide the GCNs’ performance, because the node feature distribution is also important here. In fact, the assumptions in Theorem 1 are quite strong. If the nodes with the same label are sampled from the same feature distribution, it means generally MLP can also have a good performance. When this assumption does not meet, the analysis will become very complex. That is why I think Figure 5 may be not enough to explain everything (but it is still interesting to see this empirical result).
> >
> > A3: Thanks for your recognition of the proposed cross-class neighborhood similarity metric. We agree that this metric cannot completely decide the GCN's performance, and the node feature distribution information also matters. In this paper, we limit our discussion to the case where node features are generally correlated to the node labels (or "nodes with the same label are sampled from the same feature distribution"). We clarify a few points about this assumption: 1) The assumption "the nodes with the same label are sampled from the same feature distribution" is commonly adopted in GCN/graph literature [1,2,3]. The distributions we assumed are conditional probability distributions, which can be formulated as  $p({\bf x}|y_c)$ for a given label $y_c$, where ${\bf x}$ denotes features (for those nodes with label $y_c$). It assumes that there exists some kind of correlation between the label and features, which is typically required for learning classification models. For example, generative models for classification aim to estimate such conditional probability distribution $p({\bf x}|y_c)$ for all $y_c$ from the given data; and 2) Even when this assumption is satisfied, it does not mean the MLP can always achieve good performance. Specifically, the distinguishability of the distributions for different classes is also important -- if the feature distributions for different classes are not sufficiently different, then samples from different classes will be mixed together in the feature space and the classification performance of MLP will not be good. The inter-class neighborhood similarity empirically describes the distinguishability between the neighborhood label distributions, which directly impact the distinguishability of the distributions for the output features from the GCN model and thus impact GCN's classification performance.
> >
> > When the assumption ``nodes with the same label are sampled from the same feature distribution'' does not hold, the analysis would be much more complicated. In such cases, other necessary assumptions are needed. However, it is a bit out of the scope for the current paper, we leave it for future work.
> >
> >
> > [1] Graph Convolution for Semi-Supervised Classification: Improved Linear Separability and Out-of-Distribution Generalization. ICML 2021.
> >
> > [2] Contextual Stochastic Block Models. NeruIPs 2018.
> >
> > [3] Adaptive Universal Generalized Pagerank Graph Neural Network. ICLR 2021
> >
> > ---
> >
> > Q4: Although Theorem 1 seems correct to me, I have a question here. Assume we have a separate node with 0 neighbors, that means the upper bound here is 0. It is obviously not true. So, how to explain this exception?
> >
> > A4: Note that the upper bound in Theorem 1 is $2 \cdot l\cdot \exp \left(-\frac{ deg(i) t^{2}}{ 2\rho^2({\bf W})  B^2 l}\right)$. Hence, for a separate node with $0$ neighbors, the upper bound is not $0$; it is $2\cdot l$, which is the largest value the upper bound can reach.
> >
> > ---
> >
> > We believe that we have responded to and addressed all your concerns with our revisions — in light of this, we hope you consider raising the score. Please let us know in case there are outstanding concerns, and if so, we will be happy to respond.

---

> > > ### Public Comment · ~Federico_Errica1 · 2023-01-04
> > > **Question about theorem 1**
> > >
> > > Dear authors,
> > >
> > > I have a question about the upper bound of Theorem 1. In the proof, you apply the Hoeffding's inequality according to the formulation one can find at [this link](https://en.wikipedia.org/wiki/Hoeffding%27s_inequality) for absolute values, considering that $a=-B$ and $b=B$. According to my understanding, the summation over all $n$ terms in the formula should add a $deg(i)$ at the **denominator**, not the numerator.
> > >
> > > If I am not mistaken, by applying [this definition](https://en.wikipedia.org/wiki/Hoeffding%27s_inequality) the derivation should read:
> > > $\mathbb{P} {\Large(|} \sum_{j \in \mathcal{N}(i)}(\mathbf{x}_j[k] - \mathbb{E}[\mathbf{x}_j[k]]) {\Large|} \geq t_1 {\Large)} \leq 2 \exp {\Large(}-\frac{2t_1^2}{\sum_j 4B^2}{\Large)} = 2 \exp {\Large(}-\frac{2t_1^2}{deg(i) 4B^2}{\Large)}$.
> > >
> > > I think that to obtain the upper bound reported in your work, i.e., $2\exp {\Large(}-\frac{deg(i) t_1^2}{2B^2}{\Large)}$, one would have to include $\frac{1}{|\mathcal{N}(i)|}$ in the left-hand side of the equation, but then the subsequent derivation would have to change. Similarly, $\frac{1}{|\mathcal{N}(i)|}$ is missing when computing the upper bound of $||\mathbf{h}_i - \mathbb{E}[\mathbf{h}_i]||_2$.
> > >
> > > Could you please comment about this? Am I missing something?
> > >
> > > By the way, I find your work extremely interesting, and it was a pleasure to read. I would like to build upon it but I seem to be stuck at this point.
> > >
> > > Many thanks,
> > > Federico

---

> > > > ### Public Comment · ~Federico_Errica1 · 2023-01-18
> > > > **Another question about Equation 4**
> > > >
> > > > Dear authors,
> > > >
> > > > I have another question about how you obtain Equation 4.
> > > >
> > > > Let us assume we have 2 classes and a single feature, and that $\frac{p}{p+q}=p_1$ and $\frac{q}{p+q}=p_2$. W.l.o.g.  consider a node $i$ of class $1$, then the average feature distribution (mean and variance) of a neighbor on average should be (according to my calculations)
> > > >
> > > > $\frac{1}{deg(i)}\big( p_1\mathcal{N}(\mu_1; 1) + p_2\mathcal{N}(\mu_2; 1)\big) = \frac{1}{deg(i)}\big(\mathcal{N}(p_1\mu_1; p^2_1) + \mathcal{N}(p_2\mu_2; p^2_2)\big)$
> > > > $ = \frac{1}{deg(i)}\big(\mathcal{N}(p_1\mu_1 + p_2\mu_2; p^2_1 + p^2_2)\big) = \big(\mathcal{N}(\frac{p_1\mu_1 + p_2\mu_2}{deg(i)}; \frac{p^2_1 + p^2_2}{deg^2(i)})\big)$
> > > >
> > > > Hence I am not able to obtain your same result for Equation 4.
> > > >
> > > > Could you please clarify how you obtain such a result?
> > > >
> > > > EDIT: since one has the sum over all neighbors of the expectation of variable $X$, I also think that $deg(i)$ cancels out. Does this make sense? Seeing the steps you used to get to Equation 4 through expectations will probably clarify my doubt once and for all.
> > > >
> > > > Many thanks, Federico

---

### Official Review · Reviewer_YLxN · 2021-11-04

**Correctness:** 3
**Technical Novelty And Significance:** 3
**Empirical Novelty And Significance:** 3
**Recommendation:** 6
**Confidence:** 4

**Main Review:**

Strength:

The authors provide some valuable arguments about the performance of GCN on heterophilous graphs and verify the claims with some empirical results.

Weakness&Advice:

1. In the last line of page 2, where is $X$ in the equation?

2. The results in table 1 is not consistent with my personal experience even with the hyperparameter tuning range provided in appendix D. Could you please provide more details about the settings for table 1?

3. Kenta’s work does not drop non-linearity in their analysis.

4.  Could you please clarify the significance of theorem 1? What is its relation with heterophily and homophily in definition 1?

5. Why do you use cosine similarity to define inter-class distance? What is its advantages over other metrics?

6. Font size in figure 5 is too small. The dark blue background make it hard to read the value on it.

7. The writing of this paper is not satisfying. I suggest the authors to cut the long paragraphs into smaller pieces, e.g. section 3.3.1, so that it is more reader friendly. It’s better to re-organize the paper, especially for the content after section 3.1.

8. Can your theoretical analysis on CSBM generalize to multiple classes or to more general graphs.

**Summary Of The Paper:**

The authors characterize the conditions and provide supporting theoretical understanding and empirical observations of the conditions that GCNs can achieve strong performance on heterophilous graphs.


**Summary Of The Review:**

The novelty and significance are OK but the writing is not satisfactory. I'll consider raising my score if the authors can address my concerns properly.

---

> ### Author Response · Authors · 2021-11-23
> **Response to Reviewer YLxN--Part 1**
>
> We appreciate the reviewer's recognition of our novelty and contribution. We respond to the comments/concerns from the reviewer as follows.
>
> ---
>
> Q1:  In the last line of page 2, where is ${\bf X}$ in the equation?
>
> A1: Thanks for pointing this out. We have adjusted it correspondingly in the revised version of the paper.
>
> ---
>
> Q2: The results in table 1 is not consistent with my personal experience even with the hyperparameter tuning range provided in appendix D. Could you please provide more details about the settings for table 1?
>
> A2: We have provided the code to reproduce GCN's results in Table 1. Specifically, we provide the code with detailed hyper-parameter settings in the following link (https://drive.google.com/file/d/17daB5ZvHEEJgVJawGTb6V-V1udLY445L/view?usp=sharing). We hope this could help clarify the confusion on results in Table 1.
>
> ---
>
> Q3: Kenta’s work does not drop non-linearity in their analysis.
>
> A3:  Thanks for pointing this out. We have adjusted it correspondingly in the revised version of the paper.
>
> ---
>
> Q4: Could you please clarify the significance of theorem 1? What is its relation with heterophily and homophily in definition 1?
>
> A4: Theorem 1 helps characterize the conditions for GCNs to achieve reasonable performance and correspondingly helps us understand the performance with heterophily and homophily. Particularly, it characterizes the output embeddings of the GCN model, which in turn impacts GCN's classification performance. More specifically, it bounds the intra-class distance for each class, i.e, the representations of all nodes with the same label are close to the expected representation of this class (demonstrated in Eq.(2)) with high probability (Eq.(3) demonstrates this argument). This is important as it ensures nodes with the same label to be likely classified into the same class. Then, to ensure that the classifier achieves strong performance, the expectations of different classes need to be distant from each other.  With Theorem 1, we can discuss GCN's performance with heterophily and homophily. More specifically, as mentioned in Observation 1, "In homophilous graphs, the neighborhood distribution of nodes with the same label (w.l.o.g $c$) can be approximately regarded as a highly skewed discrete $\mathcal{D}_{c}$", i.e, homophilous graphs always (approximately) satisfy the neighborhood distribution condition in Theorem 1. Furthermore, different labels clearly have distinct distributions. These make GCN models typically achieve strong performance on homophilous graphs. On the other hand, as discussed in Observation 2, for heterophilous graphs satisfying the assumptions in Theorem 1, the GCN model is also able to achieve strong performance given that the distributions of different classes are ensured to be distinguishable. In summary, Theorem 1 helps characterize the existence of ``good'' heterophilous graphs where GCN can achieve strong performance.
>
> ---
>
> Q5: Why do you use cosine similarity to define inter-class distance? What is its advantages over other metrics?
>
> A5: We utilize cosine similarity as it provides a simple, effective, and symmetric way to measure the similarity between two vectors. Other metrics can also be adopted. The choice of similarity measure would not affect the results and conclusions significantly. We empirically demonstrate this argument by investigating two other metrics: Euclidean distance and Hellinger distance. The heatmaps based on these two metrics for the Chameleon dataset are shown in Figure 11 in Appendix G. The patterns demonstrated in these two heatmaps are similar to those observed in Figure 5(b), where cosine similarity is adopted.  Note that a larger distance means lower similarity. Hence, the numbers in Figure 11 should be interpreted in the opposite way as those in Figure 5 (b).
>
> ---
>
> Q6: Font size in figure 5 is too small. The dark blue background make it hard to read the value on it.
>
> A6: Thanks for the suggestion. We have enlarged the font size. Also, we change the font color to white for those numbers in the cells with dark blue background.
>
>
> ---
>
> Q7: The writing of this paper is not satisfying. I suggest the authors to cut the long paragraphs into smaller pieces, e.g. section 3.3.1, so that it is more reader-friendly. It’s better to re-organize the paper, especially for the content after section 3.1.
>
> A7: Thanks for the suggestion. We have cut Section 3.3.1 into two separated parts. We also reorganized Section 3.2 a bit by separating long paragraphs into pieces.  In both cases, we introduced some subheadings to preface the relevant discussions and break up the text. The updates are highlighted in blue in the revision.

---

> > ### Author Response · Authors · 2021-11-23
> > **Response to Reviewer YLxN--Part 2**
> >
> > Q8: Can your theoretical analysis on CSBM generalize to multiple classes or to more general graphs.
> >
> > A8: Yes -- we extended the analysis on the CSBM for multiple classes -- please check the detailed description in Appendix F. We also believe that it is possible to generalize the analysis to more general graphs. However, the analysis will be more complicated with more factors to be considered. We think such an extension is valuable, but out of the scope of this paper. Thus, we leave this investigation for future work.
> >
> > ---
> >
> > We believe that we have responded to and addressed all your concerns with our revisions — in light of this, we hope you consider raising your score.  Please let us know in case there are outstanding concerns, and if so, we will be happy to respond.

---

### Public Comment · ~Susheel_Suresh1 · 2021-11-13
**Related work on GNNs and mixing patterns**

Hi, I wanted to bring to authors notice a related work from KDD '21 [1] which is missed in the paper and discussions. The referenced paper also analyses the behavior of GNNs w.r.t mixing patterns in networks using the notion of local assortativity. While the analysis is experimental in nature I still think it is very relevant here. Moreover, some of the conclusions obtained by the authors of the ICLR submission have also been witnessed in [1].

Wanted your thoughts on the experimental observations witnessed in [1] in relation to the current submission.

[1] Susheel Suresh, Vinith Budde, Jennifer Neville, Pan Li, and Jianzhu Ma. Breaking the Limit of Graph Neural Networks by Improving the Assortativity of Graphs with Local Mixing Patterns. KDD '21

Thanks

---

> ### Author Response · Authors · 2021-11-23
> **Response to Susheel's comment**
>
> Hi Susheel,
>
> Thanks for bringing this related work. We have included and discussed it in our revision. In the KDD'21 work, the authors found that real-world graphs typically have very diverse local assortativity and GNNs generally work better for nodes with high local assortativity. Hence, to break this limit, the authors construct a new computational graph based on structural similarity (following [2]) and the proximity in the original graph. The generated computational graph is shown to have generally higher local assortativity. The authors then proposed a new GNN framework on this graph, which improves the classification performance. Overall, the KDD'21 work aims to design new frameworks to achieve stronger performance. Our work stands from a very different perspective.  We demonstrate the existence of the "good" heterophily where GCNs can achieve reasonable performance. Our work carefully characterizes the implications of different heterophily conditions and provides supporting theoretical understanding and empirical observations.
>
> Regards,
> ICLR 2022 Conference Paper4711 Authors

---

### Decision · Program_Chairs · 2022-01-20

**Decision:**

Accept (Poster)

**Comment:**

Heterophily is known to degrade the performance of graph neural networks. This paper explores whether, for graph convolutional networks (GCNs), this is a general phenomenon, or if there are some circumstances under which a GCN can still perform well in a heterophilous setting. This paper characterizes one such setting under a contextual stochastic block model (CSBM) distribution with two classes (generalized in the appendix to multiple classes). The main takeaway is that there are indeed scenarios where a GCN can be expected to perform well, even under heterophilic neighborhoods.

There are limitations, and the reviewers have been fairly thorough in pointing these out: the analysis is specific to GCNs under CSBM, and there are a number of assumptions on the node label/feature/neighborhood distributions. The non-linear operations in the GCN have also been dropped. Even still, the reviewers were generally satisfied that the experiments backed up the claims in this specific scenario.

There is still quite a bit more to do in order to make this a more general result. Essentially, this paper shows that heterophily is not always a problem. One reviewer has stated that it is not always considered a problem anyway, but at least this paper outlines a specific scenario in which this is theoretically true. However, there is still a large space of “bad” heterophily, and this paper leaves open what these are, and how to deal with them. It is also possible that there are other “good” scenarios as well that are unexplored.

Still, in the narrow scope under which the analysis lies, the paper is clear and accomplishes what it sets out to do. I would encourage the authors to ensure that the paper incorporates the suggestions of the reviewers, particularly with regard to scope, to ensure that the paper is properly grounded in its claims.

All reviewers leaned towards the side of acceptance, except one who did not engage in post-review discussion. After reading over their review, and the subsequent response, I am satisfied that their concerns have been adequately addressed.